# Linking sardine recruitment in coastal areas to ocean currents using surface drifters and HF radar. A case study in the Gulf of Manfredonia, Adriatic Sea.

Roberta Sciascia[1], Maristella Berta[1], Daniel F. Carlson[1,6,7], Annalisa Griffa[1], Monica Panfili[1], Mario La Mesa[1], Lorenzo Corgnati[1], Carlo Mantovani[1], Elisa Domenella[1], Erick Fredj[3], Marcello G. Magaldi[1,2], Raffaele D'Adamo[1], Gianfranco Pazienza[1], Enrico Zambianchi[1,4], and Pierre-Marie Poulain[5]

[1]Istituto Scienze Marine (ISMAR), Consiglio Nazionale delle Ricerche (CNR), Italy
[2]Johns Hopkins University, Department of Earth and Planetary Science, Baltimore, MD, USA
[3]Department of Computer Sciences, Jerusalem College of Technology, Jerusalem, Israel
[4]DiST, Università degli Studi di Napoli "Parthenope" and CoNISMa, Napoli, Italy
[5]Istituto Nazionale di Oceanografia e di Geofisica Sperimentale (OGS), Trieste, Italy
[6]Department of Earth, Ocean, and Atmospheric Science, Florida State University, Tallahassee FL, USA
[7]Arctic Research Centre, Department of Bioscience, Aarhus University, Aarhus Denmark
[8]Istituto per le Risorse Biologiche e le Biotecnologie Marine (IRBIM), Consiglio Nazionale delle Ricerche (CNR), Italy

**Correspondence:** R.Sciascia (roberta.sciascia@sp.ismar.cnr.it)

**Abstract.** Understanding the role of ocean currents in the recruitment of commercially and ecologically important fish is an important step toward developing sustainable resource management guidelines. To this end, we attempt to elucidate the role of surface ocean transport in supplying recruits of European sardine (*Sardinus pilchardus*) to the Gulf of Manfredonia, a known recruitment area in the Adriatic Sea. Sardine early life history stages (ELHS) were collected during two cruises to provide observational estimates of their age-size relationship, and of their passive pelagic larval duration (PPLD). We combine these PPLDs with observations of surface ocean currents to test two hypotheses: 1) ELHS are transported from remote spawning areas (SAs) by ocean currents to the Gulf of Manfredonia; 2) sardines spawn locally and ELHS are retained by eddies. A historical surface drifter database is used to test hypothesis 1. Hypothesis 2 is tested by estimating residence times in the Gulf of Manfredonia using surface drifters and virtual particles trajectories that were computed from high resolution observations of surface currents measured by a High Frequency (HF) radar network. Transport to the Gulf of Manfredonia from remote SAs seems more likely than local spawning and retention given a mismatch between observed PPLDs of 30-50 days and relatively short ($< 10$ days) average residence times. The number and strength of connections between the Gulf and remote SAs exhibit a strong dependence on PPLD. For PPLDs of 20 days or less, the Gulf is connected to SAs on the Western Adriatic coast through transport in the West Adriatic Current (WAC). SAs on the East coast are more important at longer PPLDs. SAs in the Northern and Central Adriatic exhibit weak connections at all PPLD ranges considered. These results agree with otolith microstructure analysis, suggesting that the arrival of larvae in the Gulf is characterized by repeated pulses from remote SAs. This is the first attempt to describe the processes related to Lagrangian connection to, and retention in, the Gulf of Manfredonia that will be complemented in the future using validated numerical ocean models and biophysical models.

## 1 Introduction

Globally, pelagic forage fish, like sardines, provide important ecosystem services through their transfer of energy between trophic levels (Essington et al., 2015) and by supporting a commercial industry valued at approximately \$17 billion USD (Pikitch et al., 2014). European sardine (*Sardina pilchardus*) represents one of the most important pelagic fish resources in the Mediterranean Sea. Sardines and anchovies are the primary target species in purse-seine and mid-water pair trawl fisheries in the Adriatic Sea, with annual catches that fluctuated between $\sim$ 21000 tonnes in 2005 and $\sim$ 79000 tonnes in 2016 (SAC-GFCM, 2016). Intense fishing pressure on sardines in the Adriatic Sea resulted in large fluctuations in catches over the last 40 years, and culminated in a collapse of the fishery in the late 1980s (Morello and Arneri, 2009; Lotze et al., 2011). Catches have since increased but the stock remains over-exploited, with biomass values above the precautionary reference point (SAC-GFCM, 2016; Lotze et al., 2011). Additionally, sardines constitute an important shared fisheries resource between countries along the Adriatic coast (Albania, Bosnia ,Croatia, Italy, Montenegro, Slovenia) making marine spatial planning (e.g., Carpi et al. 2017; Depellegrin et al. 2017) particularly difficult.

In addition to fishing pressure, environmental conditions (e.g., temperature, salinity, and ocean currents) also influence the stock variability of short-lived pelagic species by impacting the survival of early life history stages (ELHS) and, therefore, recruitment strength (Peterson and Wroblewski, 1984; Bradford, 1992; Bakun, 1996; Regner, 1996; Coombs et al., 2003; Santojanni et al., 2006; Garrido et al., 2017). In general, oceanographic processes, and their modulation by high and low frequency variability, are fundamental in driving dispersal and retention of ELHS and can affect the spawning habitat and behaviour of adults, as well as the survival of eggs and larvae, largely contributing to recruitment variability (Lasker, 1981; Boehlert and Mundy, 1994; Govoni and Pietrafesa, 1994; Sabatés and Olivar, 1996; Hare et al., 2002; Sanchez-Velasco et al., 2002; Santos et al., 2004, 2018). In particular, ocean currents and their spatio-temporal variability can impact sardine recruitment during the dispersal stage, when eggs and developing larvae can be treated, at least to some degree, as passive (Largier, 2003). Thus, identifying transport pathways from spawning areas (SAs) to recruitment areas is a necessary step towards understanding complex physical-biological interactions.

The Gulf of Manfredonia is a well-known sardine nursery area that is located in the Southwestern Adriatic Sea (Fig. 1), which was historically supported a traditional trawl fishery ('Bianchetto fry' fishery) exploiting the high availability of sardine ELHS during certain periods of the year (Ungaro et al., 1994; Morello and Arneri, 2009; Carpi et al., 2016). This fishery was banned in 2010 following European regulation No. 1967/2006. Additional and more focused management actions in the Gulf would require specific ecosystem knowledge, but despite the economic and ecological importance of the Gulf many processes are still only partially understood (Specchiulli et al., 2016). For example, the mechanisms of sardine recruitment in the Gulf are not clear yet, i.e. there is no factual evidence that suggests that the nursery is supported by local spawning or by remote spawning areas. This is a relevant question, since different scenarios could call for different management strategies (Fogarty and Botsford, 2007).

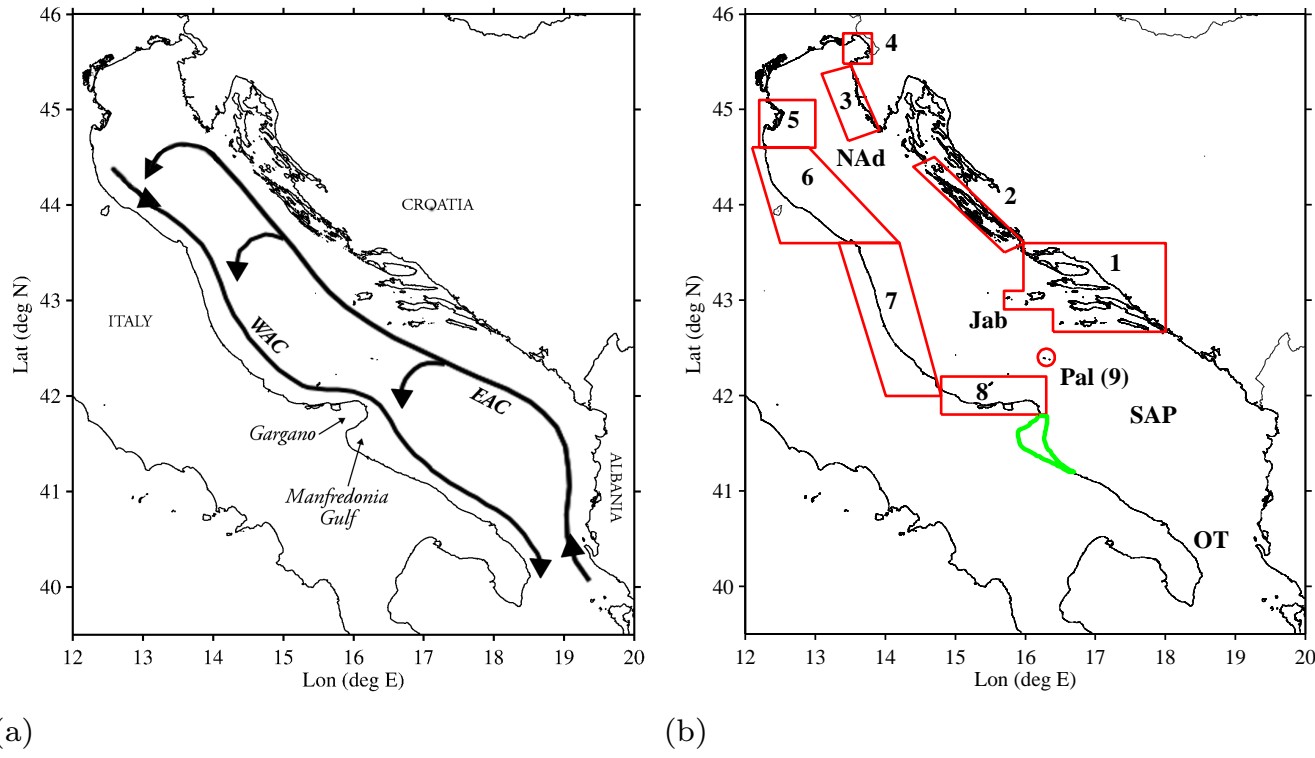

**Figure 1.** (a) The Adriatic Sea with main circulation branches: Eastern Adriatic Current (EAC), Western Adriatic Current (WAC) and the Norther, Central and Southern sub-basin gyres ( adapted from Poulain and Cushman-Roisin 2001). (b) The Adriatic Sea with the Northern Adriatic (NAD), Jabuka Pit (Jab), Southern Adriatic pit (SAP), and the Otranto Strait (OT) labeled. The SAs (outlined in red) are (1) the Southern Dalmatian Islands, (2) the Northern Dalmatian Islands, (3) the Istrian Peninsula, (4) the Gulf of Trieste, (5) the Po River Delta, (6) the Northern Italian Adriatic coast, (7) the central Italian Adriatic coast, (8) the Northern Gargano Promontory, and (9) Palagruža islands. The Gulf of Manfredonia nursery area is shown in green.

In this paper we contribute to the understanding of the Gulf recruitment processes by investigating the role of ocean current transport pathways. We focus on eggs and initial larval stages during which ELHS can be assumed to be passively advected by the currents, and we estimate the corresponding passive pelagic larval duration (PPLD; Shanks 2009) based on the analysis of sardine ELHS specimens collected in the Gulf of Manfredonia. Passive transport is then computed for different PPLD using ocean current data, with the goal of testing the following two hypotheses : 1) ELHS are remotely spawned and transported from remote SAs by ocean currents to the Gulf of Manfredonia; 2) sardines spawn locally and ELHS are retained by eddies and other small-scale retentive circulation features. The historical surface drifter database in the Adriatic Sea (Poulain et al., 2013) that provides direct measurements of transport pathways is used to test hypothesis 1. Hypothesis 2 is tested by estimating residence

times of drifters and of virtual particles trajectories computed from High Frequency (HF) radar observations of surface currents in the Gulf of Manfredonia (Corgnati et al., 2018) during the sardine spawning and recruitment season (September-May).

This is the first study that addresses the impact of ocean currents on recruitment in the Gulf using information based on actual current measurement data. Recent studies made inferences about circulation in the Gulf from relatively coarse-resolution (1/16°) ocean models (Bray et al., 2017; Specchiulli et al., 2016), but did not directly measure currents. High-resolution numerical simulations in the vicinity of the Gulf have largely focused on instabilities of the buoyant boundary current and not specifically on the dynamics within the Gulf (Burrage et al., 2009; Magaldi et al., 2010), while a first preliminary study on large-scale dispersal of eggs and larvae has been performed using velocity fields from a high-resolution ROMS model of the Adriatic Sea (Gramolini et al., 2010).

The rest of the paper is organised as follows. Section 2 summarizes the surface circulation of the Adriatic Sea, physical setting of the Gulf of Manfredonia, and the sardine population in the Adriatic Sea. The datasets and methods used to investigate recruitment dynamics in the Gulf of Manfredonia are presented in section 3. Results are presented and discussed in section 4. We provide a summary and conclusion in Section 5.

## 2   Background

### 2.1   The general circulation of the Adriatic Sea

The Adriatic Sea is an elongated (800 km by 200 km), semi-enclosed basin of the Mediterranean Sea connected to the Ionian Sea through the Strait of Otranto (Poulain, 1999). The large-scale circulation in the Adriatic Sea is predominantly cyclonic (Fig. 1a), with a Northwestward flow on the East (Balkan) coast, the Eastern Adriatic Current (EAC, typical average speed $\sim 20 - 30$ cm/s), and a Southeastward flow on the West (Italian) coast, the Western Adriatic Current (WAC, typical average speed $\sim 20 - 30$ cm/s) (Poulain, 1999, 2001; Veneziani et al., 2007; Burrage et al., 2009). These boundary currents are responsible for swift alongshore transport. In addition to the basin-scale cyclonic circulation smaller and persistent cyclonic recirculations (Fig. 1a) have been observed in the Northern, central, and Southern Adriatic (Poulain, 1999; Burchard et al., 2008). Previous surface drifter and modeling studies have shown that preferential cross-basin exchange can occur in these sub-basin gyres (Poulain, 2001; Carlson et al., 2016, 2017).

Circulation in the Adriatic Sea is driven by buoyancy and winds and is constrained by topography (Orlić et al., 1994; Poulain, 1999; Burrage et al., 2009). In particular the WAC is a buoyant coastal current driven largely by the discharge of the Po River in the Northern Adriatic. Both the EAC and the WAC are significantly modulated by the winds. Southeasterly and Northwesterly winds blow parallel to the coast and, if persistent, can drive upwelling or downwelling (Magaldi et al., 2010). Northwesterly winds, most common in summer (Pasarić et al., 2009; Magaldi et al., 2010), are downwelling-favorable along the Eastern Italian coast and tend to strengthen the WAC, increasing its vertical thickness and suppressing the formation of instabilities, while weakening the EAC (Burrage et al., 2009; Magaldi et al., 2010). Opposite dynamical effects occur with Southeasterly winds (Orlić et al., 1994; Poulain et al., 2004; Pasarić et al., 2007; Magaldi et al., 2010)

## 2.2 The Gulf of Manfredonia

The Gulf of Manfredonia is located on the Western Adriatic coast (Fig. 1) in the transition zone from the North-central to the Southern Adriatic Sea (Damiani et al., 1988; Spagnoli et al., 2004; Balestra et al., 2008; Focardi et al., 2009). The Gulf is delimited by the Gargano Promontory to the North and the curvature of the coastline to the South. The Gulf differs from the rest of the central and Southwestern Adriatic coast due its shallow depths, gently sloping bottom, abrupt coastline curvature, variable circulation, and eutrophic waters (Focardi et al., 2009; Campanelli et al., 2013; Marini et al., 2015). The Gulf is a productive area that supports commercial fisheries for several species (Vaccarella et al., 1998; Grilli and Falcone, 2010; Borme et al., 2013), serves as a nursery area for sardines and anchovies (Morello and Arneri, 2009; Borme et al., 2013), and is a foraging area for sea turtles (Casale et al., 2012; Casale and Simone, 2017). It supports a local commercial fishing industry that is in decline, presumably due to over-fishing. Aside from over-fishing, anthropogenic activities in the form of industrial and urban waste, and maritime shipping traffic threaten the ecosystem and economy of the Gulf (Damiani et al., 1988; Accornero et al., 2004; Focardi et al., 2009; Garcia et al., 2013; Monticelli et al., 2014; Suaria and Aliani, 2014).

Hydrographic observations have separated the Gulf into three zones: a well-mixed, shallow nearshore zone, a transition zone characterized by weak stratification, and an Eastern, offshore stratified zone (Balestra et al., 2008; Focardi et al., 2009; Monticelli et al., 2014). The nearshore zone extends to approximately the 15 m isobath and is characterized by lower (higher) temperature in summer (fall/winter) than offshore waters, while surface salinities are consistently lower nearshore throughout the year, presumably due to river discharge (Balestra et al., 2008; Focardi et al., 2009; Campanelli et al., 2013). The transition zone is located between the 15-25 m isobaths and is weakly stratified (Focardi et al., 2009). The offshore, stratified zone begins at the 25 m isobath where a thermohaline front typically separates inner Gulf waters from denser offshore Adriatic Sea waters (Balestra et al., 2008; Focardi et al., 2009; Monticelli et al., 2014).

In winter, temperature in the Gulf is approximately 10°C (Casale et al., 2012; Borme et al., 2013). Surface temperatures vary from 14-18.5°C in April - May (Campanelli et al., 2013; Casale et al., 2012; Monticelli et al., 2014), increase to over 23°C in June (Focardi et al., 2009; Casale et al., 2012) and decrease to approximately 19°C in October (Balestra et al., 2008). Surface salinities range from 35.2 - 38 in April - May (Campanelli et al., 2013; Monticelli et al., 2014), 37-38 in June (Focardi et al., 2009), and 37.5 in October (Balestra et al., 2008).

The effects of wind, WAC intrusions and local river discharge influence water properties in the Gulf (Balestra et al., 2008; Campanelli et al., 2013; Specchiulli et al., 2016). The largest local river is the Ofanto (average discharge of 13.9 $m^3s^{-1}$) with other, smaller rivers (Candelaro and Cervaro) active during the winter rainy season and practically dry in summer (Damiani et al., 1988; Spagnoli et al., 2008; Focardi et al., 2009; Infante et al., 2012; Campanelli et al., 2013; Specchiulli et al., 2016). Vertical sections of temperature and salinity suggest the occasional presence of estuarine circulation in the Gulf, with a high degree of variability in space and time (Focardi et al., 2009). Nutrients, colored dissolved organic matter (CDOM), chlorophyll-a, fluorescence, and turbidity have their highest values nearshore, likely due to river discharge (Focardi et al., 2009; Campanelli et al., 2013), but they are subject to high variability especially regarding the exchange with offshore waters induced by physical forcing.

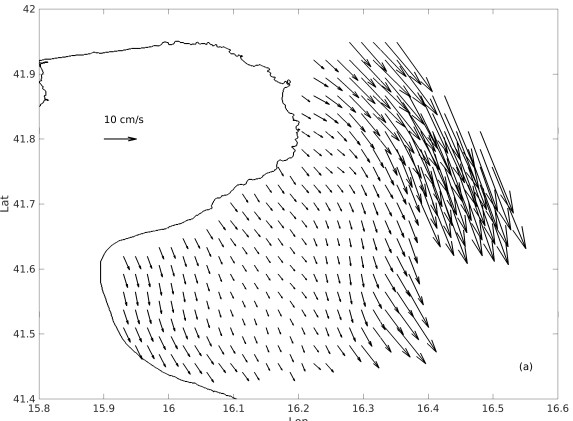 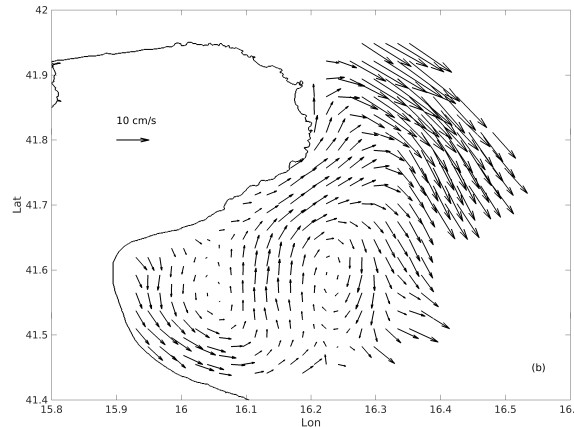

**Figure 2.** Mean surface currents in the Gulf of Manfredonia measured by the ISMAR HFR network. (a) October 2013: currents are dominated by a large-scale pattern, (b) February 2014: many different small-scale structure are visible.

Currents at the outer edge of the Gulf are affected by the energetic WAC and its instabilities, while currents in the inner Gulf are influenced by winds (Focardi et al., 2009; Campanelli et al., 2013) as well as by the variability of the large scale current (Specchiulli et al., 2016). Multiple studies (Spagnoli et al., 2004; Balestra et al., 2008; Spagnoli et al., 2008; Focardi et al., 2009; Campanelli et al., 2013; Monticelli et al., 2014) state that the circulation in the Gulf is weak and dominated by cyclonic and anticyclonic gyres that form in response to the local winds. However, recent results from HF radar surface current measurements (Corgnati et al., 2018), showed a velocity field with typical values of the order of 10-20 cms$^{-1}$, often reaching peaks of $\sim$ 30 cms$^{-1}$, and characterised by a richness of small-scale structures and significant variability in both time and space. Examples of monthly mean surface velocity variability in the GoM from HF radar are shown in Fig.2. The October flow is dominated by a large scale pattern and has low (less than 10cm/s) mean velocities, while the February currents show many structures at different scales with higher mean velocities.

## 2.3 Sardines in the Adriatic Sea

The life cycle of sardines in the Adriatic Sea were recently reviewed by Morello and Arneri (2009) and here we summarize the most pertinent information for the present study (for a detailed description see Morello and Arneri 2009 and references therein). Sardines feed primarily on zooplankton and are preyed upon by larger predators, thereby forming an important link from lower to higher trophic levels in the Adriatic Sea, and elsewhere (e.g., Nikolioudakis et al. 2011; Borme et al. 2013; Costalago 2015; Costalago et al. 2015). Therefore, the reduction of sardine stock in the Adriatic Sea can impact the entire ecosystem (Morello

and Arneri, 2009; Carpi et al., 2017). After the abrupt decline in the late 1980s, the sardine population increased in the late 1990s but quickly resumed its downward trend, with a minimum historical catch reported in 2005 (Morello and Arneri, 2009).

Systematic ichthyoplanktonic surveys targeting small pelagic fishes were carried out between 1976 and 1996 by the Laboratorio di Biologia Marina in Fano (Italy) in collaboration with the University of Trieste (Italy) and the Institute of Oceanography and Fisheries in Split (Croatia) in order to describe the spawning habitat and the spawning distribution of sardines (Piccinetti et al., 1980, 1981; Regner et al., 1981, 1987, 1988). After 1996, surveys were conducted only in a smaller area near the Po River mouth (Coombs et al., 2003).

In the Adriatic Sea sardines spawn between September and May, with at least one or two reproductive peaks. The timing and location of spawning change in relation to local environmental factors (mainly temperature and salinity) and food availability. Adult sardines tend to avoid extremes in temperature and salinity and migrate in search of optimal environmental conditions and sufficient food (Palomera et al., 2007; Morello and Arneri, 2009). The presence of eggs has been observed over a wide area of the Adriatic continental shelf. On both flanks of the basin, the Northern Adriatic and Central Adriatic up to the Gargano promontory to the West and the Southern Dalmatian Islands to the East, have been recognized as the most important and intense areas for spawning (Morello and Arneri, 2009). Only in some years, the Southern portion of the basin may be affected by spawning events extending along the Italian coast down to Otranto (e.g. Piccinetti et al. 1981; Gamulin and Hure 1983).

For this reason, nine spawning areas (SAs) in the northern and central Adriatic are considered here and are outlined in Figure 1b. The SAs include: Southern Dalmatian Islands (1), Northern Dalmatian Islands (2), Istrian Peninsula (3), Gulf of Trieste (4), Po River Delta (5) , Northern Italian coastline (6), the Central Italian coastline (7), the Northern Gargano Promontory (8), and the Palagruža Islands (9) (Piccinetti et al., 1980, 1981; Regner et al., 1981, 1987; Sinovčić and Alegriahernandez, 1997; Sinovčić, 2001, 2003). The Gulf of Manfredonia has been typically described as a nursery area fostering larvae and juveniles spawned elsewhere (Morello and Arneri, 2009). Ichthyoplanktonic surveys showed great abundance of sardine post-larvae and juveniles, whereas the number of eggs found in the Gulf was low (Panfili, 2012; Borme et al., 2013). In fact, environmental conditions in the Gulf of Manfredonia might not be suitable during the winter reproductive months, while they are suitable for nursery during the following months. In particular due to river discharge and rains, the waters inside the Gulf are fresher than the waters in other areas of the Adriatic (Morello and Arneri, 2009).

During the early ontogeny of sardine, a series of development stages characterized by key morphological and functional changes takes place. The transition from the larval to the juvenile stage and, consequently, the remodeling of the organism often involves substantial changes that represent a crucial time for individual larvae and for recruitment success. Commonly, in optimal environmental conditions, larvae develop quickly until they reach the juvenile stage after a gradual or abrupt process called metamorphosis, by which larvae acquire most morphological and physiological characteristics of adults including complete ossification, scale formation, full set of fin rays, etc. The definition of size, timing and morphological criteria determining the transition from passive transport to active swimming in the water column are rather arbitrary and have not been clearly defined yet, because they are strictly dependent on local temperature and food intake. High temperatures can accelerate the timing of ontogenetic development more than the rate of growth (Fuiman et al., 1998). Not only do larvae grow faster at higher temperatures, but ontogenetic development might occur sooner and at smaller size (Garrido et al., 2016).

In this study, in agreement with Santos et al. (2007) and Brochier et al. (2008), the transition from passive transport to active swimming in sardine larvae was considered at the onset of pelvic fin formation, taking place approximately at 20 mm total length (TL). According to the linear growth model recently applied to sardine larvae collected in the Western Adriatic Sea (Panfili, 2012; Domenella et al., 2016), this size corresponds to an age estimate of 30-40 days. However, given the uncertainties in determining the transition to active swimming, we will consider a wider PPDL range up to 60 days from larval hatching, as discussed in Section 3.4 and 4.1.

## 3  Data and Methods

### 3.1  Early life stages data and analyses in the Gulf of Manfredonia

ELHS of sardines were collected in coastal waters within the Gulf of Manfredonia during two cruises carried out in winter during the peak of the spawning season (March 2013 and February 2014). Fish samples were used to establish the relationship between size and age and to provide estimates of PPLD.

Samples were collected using a pelagic trawl net equipped with a fine-meshed code-end (mesh size of 5 mm, horizontal net opening = 3.7 m, vertical net opening = 2.1 m). Hauls were carried out along inshore-offshore transects for approximately 30 minutes at an average speed of 3.0 knots (1.54 ms$^{-1}$) to avoid cod-end stressing of fish. Sampling depth varied between 5 and 50 m depending on the area. After the end of each haul, all individuals were sorted from the trawl catch and immediately preserved at $-20°$C. In the laboratory, all specimens were examined to assess their larval stage and their total length was measured rounded to the nearest millimeter. Otolith microstructure analysis and daily growth increment counts were performed to assess ageing structure and growth rate of the local population, as well as to provide an estimate of larval durations before metamorphosis (PPLD).

Sagittal otoliths were extracted from each individual and processed according to Domenella et al. (2016). Otolith pairs were randomly selected and mounted in epoxy resin (Petropoxy) on a glass slide (medial side down) and were polished using fine grit lapping film (1-3 $\mu m$ and then 0.05 $\mu m$) to enhance the pattern of daily increments. The daily increments were counted from the nucleus to the edge along the longer axis under a light microscope at $400\times$ magnification equipped with a CCD camera that was connected to an image analysis system (Image-Pro Plus 7). Two increment counts were made for each otolith and and counts that differed by more than 10% from the mean value were discarded.

All the individual values of TL and age were used to quantify their functional relationship. Due to the relatively narrow fish size range, a linear model has been assumed appropriate to estimate the daily growth rate of the sampled population (Campana and Jones, 1992).

### 3.2  Surface drifter Data

Drifters are floating buoys that follow marine currents with good approximation, providing direct information on velocity and transport (see, e.g., Lumpkin et al. 2017). They have been previously used to study connectivity in several areas of the world

ocean (Brink et al., 2003; Condie et al., 2005; Cowen et al., 2000; Gawarkiewicz et al., 2007; Carlson et al., 2016). The data used in this work consist of CODE-type (Coastal Ocean Dynamics Experiment) drifters (Davis et al., 1981; Davis, 1985) that sample the ocean surface in the first meter of water. Dedicated experiments with current meters show that CODE drifters are consistent with the near-surface Ekman dynamics and have reduced errors of the order of 1-3 cm/sec for wind regimes up to $\sim$ 10 m/sec (Davis, 1985; Poulain et al., 2009). For these reasons, CODE drifters have been extensively used in the literature to describe mesoscale and submesoscale transport of passive tracers at the ocean surface (Berta et al., 2014, 2016).

The data set used here includes a total of 393 drifters launched in the Adriatic Sea during the period 1994-2015 and covering the whole basin (Poulain, 1999, 2001; Poulain and Hariri, 2013; Poulain et al., 2013). In particular, the data set includes 26 drifters released in the central Adriatic in May 2013 during the CoCoPro experiment (Boero et al., 2016) to investigate connections and transit times between marine protected areas (MPAs, Carlson et al. 2016), as well as 5 drifters released in the Gulf of Manfredonia in February 2014 as part of validation tests of the HF radar velocities (Corgnati et al., 2018). The dataset has been quality controlled to remove spikes and offsets, and the drifters positions have been low-pass filtered and subsampled at 6 hr intervals (see Poulain 1999, 2001; Poulain and Hariri 2013). Note that part of the dataset, namely the 5 Manfredonia drifters, were released during the same sampling period of the ELHS of sardines (see Section 3.1) and of the HF radar velocity measurements (see Section 3.3).

### 3.3 HF radar data

HF radars provide maps of ocean currents over extensive areas of the coastal ocean with time intervals of the order of one hour (for a recent review of their characteristics see Paduan and Washburn 2013). They have reached quite a wide distribution over the ocean coastlines (Rubio et al., 2017); their high-temporal and spatial resolution and their synoptic view allow for a continuous and detailed monitoring of coastal dynamics and of transport processes occurring in coastal areas (Bellomo et al., 2015). The regularity of their data coverage in space and time, and the possibility of processing and disseminating such data in near-real time makes them extremely valuable in the framework of coastal oceanography both for operational (Falco et al., 2016; Iermano et al., 2016) and for ecological (Helbig and Pepin, 2002; Bassin et al., 2005; Morgan et al., 2012; Cianelli et al., 2017) studies.

A HF coastal radar network was installed and maintained during the period August 2013 - June 2015 in the Gulf of Manfredonia by ISMAR-CNR. The network was composed of four CODAR SeaSonde direction finding systems operating at 25 MHz, installed in four sites within the Gulf with the best available spacing in order to cover its interior. The locations of the four sites (Vieste, Pugnochiuso, Mattinatella, and Manfredonia) are shown in Fig. 3. Under the framework of the JERICO-NEXT project the raw data are harmonised and combined into a geographical grid with a spatial resolution of 1.5 km and provided every hour (Corgnati et al., 2018).

Similarly to Kalampokis et al. (2016), HF radar surface velocities in the Gulf ware validated using in-situ measurements from drifters (Corgnati et al., 2018). The results show a good agreement, keeping into account the different nature of the two platforms and the environmental variability unresolved by the radar within the 1.5 km grid. The *rms* (root mean square) of the differences between drifter and HF radar velocities is $\sim 20\%$ - $50\%$ of the drifter *rms* velocities, which falls on the lower side

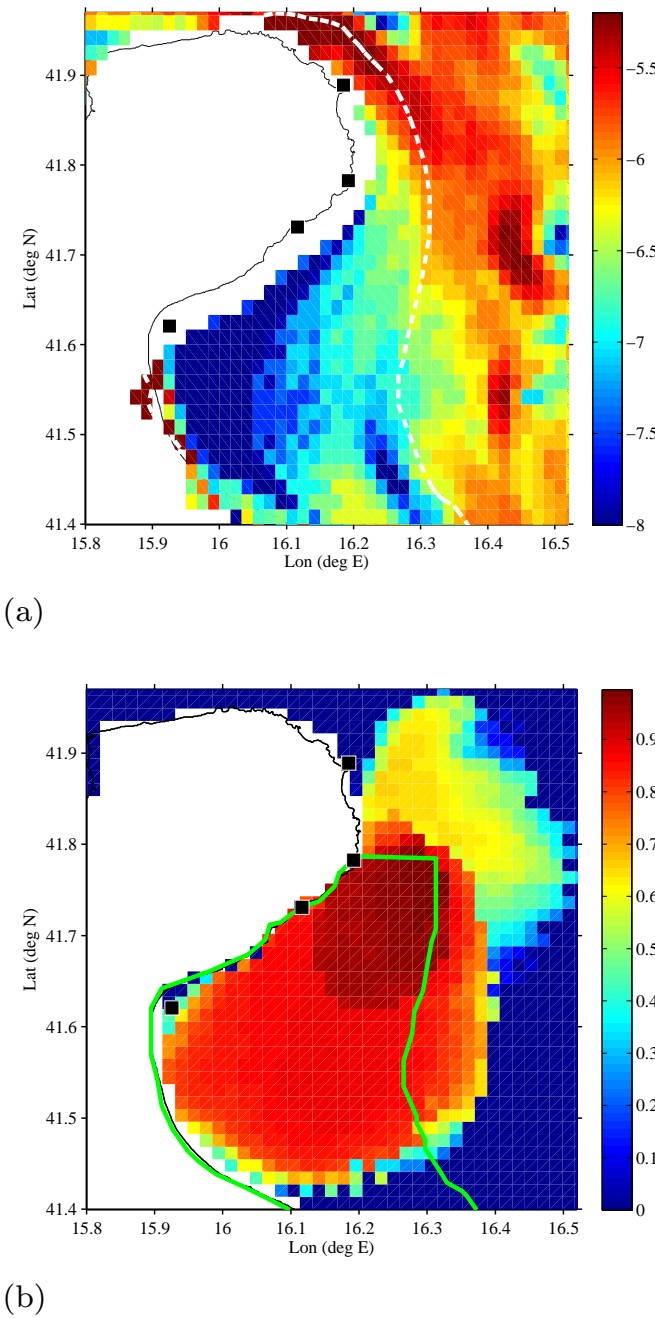

**Figure 3.** (a) The Gulf of Manfredonia with the four HF radar installations denoted by black squares: from North to South Vieste, Pugnochiuso, Mattinatella, and Manfredonia, respectively. Colors indicate the logarithm of the bottom slope, and the dashed white line corresponds to the 25 m isobath used to define the offshore boundary of the Gulf. (b) The average HF radar radial coverage and the relative data density during the period October 2013 - May 2014. The boundaries of the Gulf of Manfredonia as defined in Section 3.4 are represented by the green line.

| Number | SA | N in SA | N Gulf | % | 95% CI |
|---|---|---|---|---|---|
| 1 | S. Dal | 69 | 6 | 8.7 | 4 - 17.7 |
| 2 | N. Dal | 46 | 3 | 6.5 | 2.2 - 17.5 |
| 3 | Istr. Penin. | 58 | 0 | 0 | 0 - 0 |
| 4 | Trieste | 7 | 0 | 0 | 0 - 0 |
| 5 | Po | 74 | 1 | 1.3 | 0.2 - 7.3 |
| 6 | N. Ital. | 114 | 9 | 7.9 | 4.2 - 14.3 |
| 7 | C. Ital. | 109 | 15 | 13.8 | 8.5 - 21.5 |
| 8 | N. Garg. | 106 | 26 | 24.5 | 17.3 - 33.5 |
| 9 | Pal. | 23 | 2 | 8.7 | 2.4 - 26.8 |

**Table 1.** Connection of SAs with the Gulf of Manfredonia estimated by drifters. Column 1 and 2 indicate the number and name of each SA (see Fig. 1b for locations). Column 3 indicates the total number (N) of drifters that entered or were launched in a given SA, while column 4 indicates the number of drifters that continued from the SA to the Gulf of Manfredonia (N Gulf). The percentage (%) of N Gulf drifters with respect to N is shown in column 5. Column 6 shows the 95% confidence intervals estimated using the Wilson Score.

of typical errors found in literature. The reader is referred to Corgnati et al. (2018) for a detailed description of the HF-Radar validation.

The HF radar footprint covers most of the Gulf, approximately 1700 km$^2$ (Fig. 3b). In this work, we focus on the internal part of the Gulf (Fig. 3a), i.e, within the 25 m isobath, (as further explained in subsection 3.4), and on the spawning period of sardines, between September and May. Good HF radar coverage within this period was achieved in October 2013 - May 2014, and the analysis focuses on this period. The corresponding average coverage is shown in Fig. 3b, and it corresponds to $\sim 80 - 90\%$ of the measurements in the area within the 25 m isobath. Spatial and temporal gaps that occasionally occur in HF radar data due to environment conditions (Kohut and Glenn, 2003; Gurgel et al., 2007; Laws et al., 2010), have been interpolated using the recently developed method of Fredj et al. (2016).

## 3.4 Connections between spawning areas and the Gulf of Manfredonia using drifters

Drifters are a natural choice to test the first hypothesis that ELHS are transported to the Gulf of Mandredonia from remote SAs by ocean currents. To this end, we use a simple method, similar to the one previously used by Carlson et al. (2016). As a first step, we define the exact geometry of the regions of interest in Figure 1b). The historically known SAs (Morello and Arneri, 2009) are characterised as the nine boxes delimited by the red lines. Each SA is associated to a number (Table 1), starting from the most Southeastern SA (Southern Dalmatian Islands), and increasing following the coast cyclonically. The Gulf (Fig.3b, green line) is assumed to be defined by the 25 m isobath as offshore Eastern boundary. This choice is consistent with previous works showing that the 25 m isobath typically separates inner Gulf waters from offshore Adriatic Sea waters (Carlson et al., 2016; Corgnati et al., 2018), and is in agreement with the steep depth gradient shown in Fig. 3a. The Southern boundary is set along 41.40 N, in order to take into account the HF radar coverage (Fig. 3b).

As a second step, for each SA we identify all the drifters that either enter the area or were launched in it. Of all these drifters, we tag the ones that reach the Gulf. The percentage of tagged drifters reaching the Gulf is computed as a simple measure of the connection between the considered SA and the Gulf. 95% confidence intervals around the connection percentage are estimated using the Wilson Score (Agresti and Coull, 1998).

    This calculation is first performed considering the complete data set to provide bulk estimates, and then considering condi-
25 tional sets in terms of seasons and PPLD. In particular, we concentrate on the sardine spawning season, September-May, and consider the following PPLD ranges: 0-20 days; 20-40 days ; 40-60 days. Notice that, as mentioned in Section 2.3 and further expanded in Section 4.1, typical PPLDs in the Adriatic are likely to be in the range of 30-40 days. Here we extend the range considering smaller and bigger values to take into account uncertainties in determining the PPLD and possible effects of environmental variability. Moreover, the analysis of extended PPLD values allows us to better understand the effects of transport
by the currents.

    Conditional sets are created considering drifters in a given SA only if they enter during the considered season and if their lifetimes after leaving the SA are at least equal to the lower limit of each PPLD interval considered. That is, if a drifter still has enough lifetime to reach the Gulf nursery area during one of the three PPLD ranges considered. Note that the average lifetime ($\sim$90 days) of drifters entering the Gulf of Manfredonia is well above the maximum PPLD considered in the analysis.

    In addition to the connection computation, we also consider some bulk statistics (Carlson et al., 2016) that provide informa-
5 tion on the density distribution of drifter trajectories before and after entering the Gulf.

### 3.5     Residence times in the Gulf from drifters and HF radar data

To test the hypothesis 2 that sardines spawn locally and ELHS are retained within the Gulf of Manfredonia, we estimate average residence times in the Gulf using drifters and HF radar velocities. Residence times are defined as the amount of time from first entry to first exit (Menna et al., 2007).

10     The number of drifters that enter the Gulf or were launched in it is relatively small, of the order of 30 drifters over the whole data set. The drifter-based residence times cannot therefore be very detailed, and drifters are mostly used to gain some qualitative insight on distribution and entering and exiting pathways.

    More quantitative statistics of residence times are computed using HF radar data. The hourly velocity fields of the radar $\mathbf{u}$ were used to compute virtual particle trajectories, $\mathbf{x}$(t), solving the equation

$$\frac{d\mathbf{x}(\mathbf{x},t)}{dt} = \mathbf{u}(\mathbf{x},t). \tag{1}$$

The Matlab ode45 solver (ode45; 4,5th order Runge-Kutta) was used to perform particle tracking, with an adaptive time step set to a maximum value of one hour consistent with the sampling frequency of the HF radar surface currents. We compute two-
5 dimensional virtual, passive particle trajectories by integrating the observed HF radar surface currents in time to approximate particle positions. Particles were seeded at the 394 HF radar grid points within the Gulf. All particles were treated as passive, buoyant tracers with no attempt to include larval behaviour or to parameterize subgrid-scale diffusion. Forward particle trajectories were computed for 30 days and reseeded every 12 hours to estimate the residence time of a buoyant, passive tracer in the

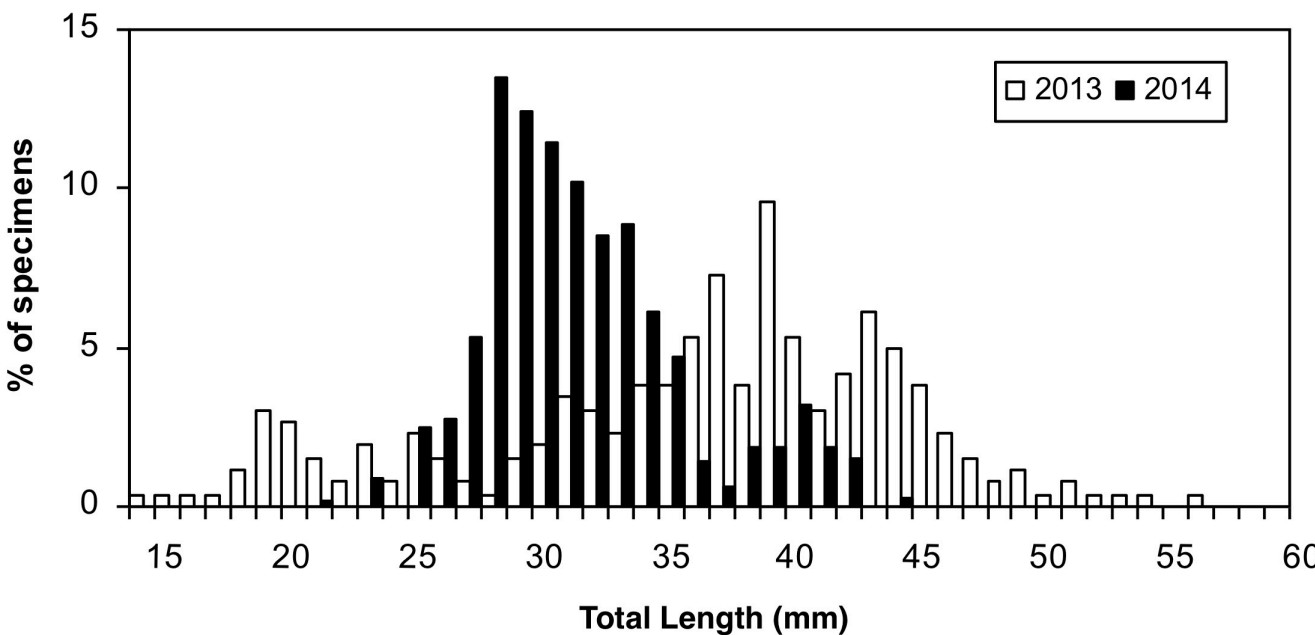

**Figure 4.** Length frequency distributions of early life stages of sardine collected in the Gulf of Manfredonia.

Gulf. Monthly mean residence times of virtual particles in the Gulf were computed using the bootstrap resampling technique (Efron and Tibshirani, 1986).

Note that residence times are computed here using the time of first exit for each virtual particle. In reality, it is possible that some of the particles re-enter the Gulf after some time spent offshore, therefore expanding the time of actual influence of the Gulf. This effect cannot be accurately determined using the radar coverage.

## 4   Results

### 4.1   Age structure and growth rate of sardine larvae

Otolith microstructure analysis conducted on 416 individuals caught in Manfredonia Gulf in two different surveys, March 2013 (n=262) and February 2014 (n=154), showed that fish ranged in size from 15 to 57 mm. TL and the relative age estimated from the TL vary from 21 to 136 days. In 2014, catches consisted mainly of late larvae before the metamorphosis (TL $<=$ 30-35 mm), while in 2013 catches also included juveniles, probably because the survey was carried out later in winter (Fig. 4). The relatively wide and bimodal/polymodal length frequency distributions suggest a continuous occurrence of newly hatched larvae and suggest a prolonged spawning season characterised by different egg pulses. The mean daily growth rate varied from 0.27 mm day$^{-1}$ to 0.23 mm day$^{-1}$ in 2013 and 2014, respectively. The hatch date distribution, back-calculated from the age

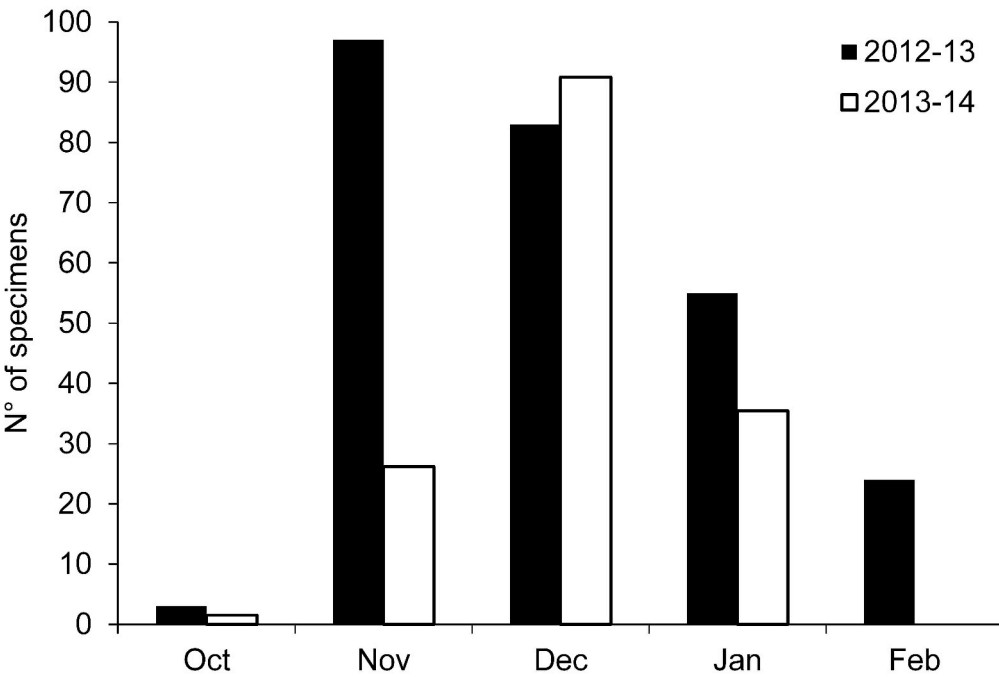

**Figure 5.** Sardine hatch date distribution back-calculated from date of catch and estimated age. Different colors represent the two sampling periods in the Gulf of Manfdredonia.

estimates and dates of capture, spreads over an extended period lasting from fall (October) to winter (February). Hatching peaks were slightly different in 2013 and 2014, occurring in November and December, respectively (Fig. 5).

Estimates of PPLD values, obtained from larval age before metamorphosis, confirm and slightly expand the range of previous estimates in the Western Adriatic by Domenella et al. (2016), suggesting a PPLD range of $\sim$ 30-40 day.

## 4.2    Connections between spawning areas and the Gulf

### 4.2.1    Results for the complete data set: unconditional statistics

A general description of the connections between the Gulf and the Adriatic Sea is first obtained using the complete drifter data
set. An overview of the main pathways of connections is provided by the maps of density distribution of drifters leaving and entering the Gulf. (Fig. 6 a,b), obtained considering fixed bins/areas of 0.25° (approximately 25 km) and counting the number of non-consecutive entries of each drifter into a given bin/area.

The highest concentrations are found along the Italian coast, indicating that the WAC plays a major role as a pathway for both entering and exiting the Gulf. Relatively large drifter concentrations can also be seen in the recirculating sub-basin gyres.

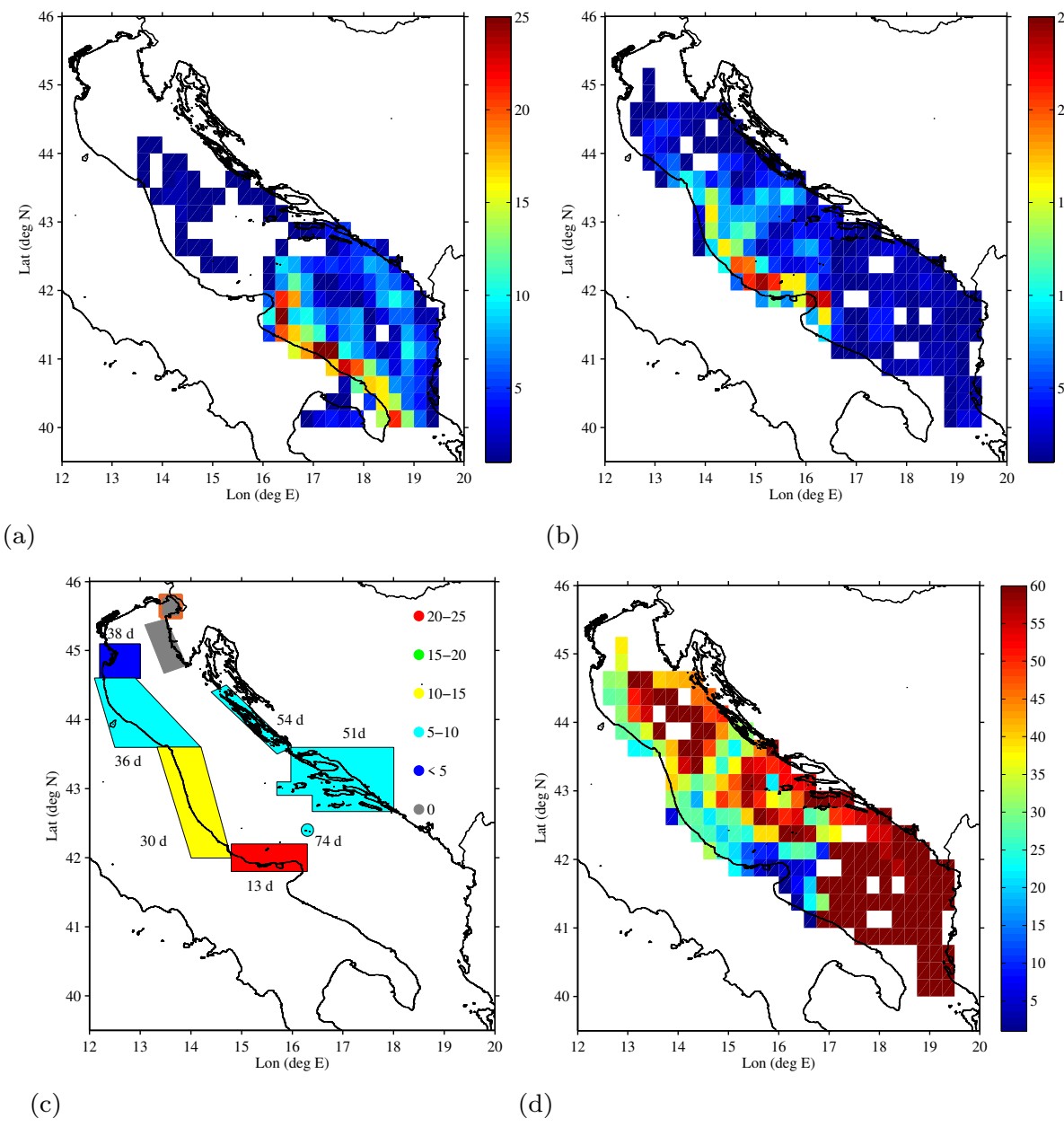

**Figure 6.** Density of drifter data from (a) trajectories that leave the Gulf of Manfredonia; (b) trajectories that enter the Gulf of Manfredonia. Densities refer to the number of nonconsecutive entries into each $0.25^o$ bin. (c) The connection percentages (i.e. percentage of drifters that passed through a given SA and reached the Gulf of Manfredonia) and the average time to reach the Gulf from each SA computed using the complete drifter data set (see Table 1). Results obtained with less than 10 drifters per SA, as for the Gulf of Trieste, are disregarded and identified by the orange contour around the SA. (d) Transit times of drifters coming from the whole Adriatic Sea entering the Gulf of Manfredonia.

Drifters exiting the Gulf (Fig. 6a) are likely to be caught in the Southern sub-basin gyre, while drifters entering the Gulf show a well defined pathway that follows the central sub-basin gyre.

The results of the connection computations for all the SAs are summarised in Table 1 and in Fig. 6c. In the following discussion, the SAs are indicated with their name and with their corresponding number (Table 1 and Fig. 1b). The number of drifters found in each SA is typically in the range 50-100, except for the case of Gulf of Trieste (SA 4) and Palagruža Islands (SA 9), that have only 7 and 23 drifters respectively. The maximum number of drifters reaching the Gulf is found near the Gargano Promontory (SA 8), with a percentage of almost $25\%$. This is not surprising given that SA 8 is adjacent to the Gulf and upstream with respect to the WAC. The connections of the other SAs show some interesting trends, that are evident despite the relatively wide confidence intervals ($95\%$). The connection percentages decrease going North along the Western coast of the Adriatic, reaching values around $1\%$ at the Po River Delta (SA 5) (Fig. 6c). This is consistent with the increasing distance from the Gulf along the WAC. Connections with SAs 1 and 2 along the East coast are relatively high ($5-10\%$). This is due to the central Adriatic sub-basin gyre, that advects particles away from the Northward EAC, crossing the Adriatic and reaching the Southward WAC. Finally, Gulf of Trieste and Istrian Peninsula (SAs 4 and 3) show zero connection. However, the Gulf of Trieste value is not significant, given the small number of drifters, but the Istrian Peninsula result is significant, and it suggests that the Northeastern SAs tend to be isolated from the Gulf of Manfredonia.

In order to gain some insight on the time scales involved, Fig.6c also indicates the average time required to reach the Gulf from each SA. Drifters leaving SA 8 need on average 13 days, while the SAs located on the western Adriatic coast need on average 35 days and SA located on the eastern coast need about 50 days. Even though average times for some SAs are not quantitatively significant given the small number of drifters reaching the Gulf (see Table 1), results are consistent with the unconditioned statistics (Fig. 6d) showing the average transit time of drifters reaching the Gulf from the whole Adriatic Sea. Most of the drifters coming from the area North of the Gulf need 10 days or less, in agreement with the southward flow of the WAC. On the other hand, drifters coming from the Northern-Central basin require more time to reach the Gulf. This is a consequence of the advection from the recirculating sub-basin gyres.

### 4.2.2 Results for the sardine spawning time period: conditional statistics

Connections between the SAs and the Gulf during the sardine spawning period are investigated in detail, selecting drifters entering the SAs during the period September-May and considering three different PPLD ranges: 0-20 days; 20-40 days; 40-60 days. The results are summarised in Table 2 and in Fig. 7.

The number of drifters in each SA is obviously smaller than for the complete data set in Table 1, and it decreases with increasing PPLD, reaching $\sim$ 20-40 for the longest PPLDs. Results obtained with N less than 10, as for Gulf of Trieste (SA 4), are disregarded and are identified in Fig. 7 using an orange contour around the SA. With decreasing N, the $95\%$ confidence intervals widens, implying that the specific percentage values for each SA becomes less certain.

Results for the lowest PPLD range, 0-20 days (Fig. 7a), show that the highest connection percentage is obtained for Gargano Promontory (SA 8), just North of the Gulf. The percentages decrease Northward along the Western Adriatic coast, similar to what is shown for the complete data set in Fig. 6c. Differently from Fig. 6c, though, all the SAs along the Eastern coast (SAs

| Number | SA | PPLD = 0-20 days | | | PPLD = 20-40 days | | | PPLD = 40-60 days | | |
|---|---|---|---|---|---|---|---|---|---|---|
| | | N in SA | N Gulf | %, [95% CI] | N in SA | N. Gulf | % | N in SA | N Gulf | % |
| 1 | S. Dal | 64 | 0 | 0 | 45 | 2 | 4.4,[1.2 - 14.8] | 32 | 2 | 6.2,[1.7 - 20.1] |
| 2 | N.Dal | 40 | 0 | 0 | 27 | 1 | 3.7,[0.7 - 18.3] | 17 | 1 | 5.9,[1.0 - 27.0] |
| 3 | Istr. Penin | 44 | 0 | 0 | 34 | 0 | 0 | 20 | 0 | 0 |
| 4 | Trieste | 2 | 0 | 0 | 1 | 0 | 0 | 1 | 0 | 0 |
| 5 | Po | 61 | 0 | 0 | 36 | 1 | 2.8,[0.5 - 14.2] | 23 | 0 | 0 |
| 6 | N. Ital. | 91 | 2 | 2.2,[0.6 - 7.7] | 55 | 4 | 7.3,[2.9 17.3] | 42 | 1 | 2.4,[0.4 - 12.3] |
| 7 | C. Ital. | 85 | 7 | 8.2,[4.0 - 16.0] | 58 | 1 | 1.7,[0.3 9.1] | 41 | 2 | 4.9,[1.3 - 16.1] |
| 8 | N. Garg. | 92 | 19 | 20.6,[13.6 - 30.0] | 57 | 0 | 0 | 43 | 2 | 4.6,[1.3 - 15.4] |
| 9 | Pal. | 23 | 0 | 0 | 19 | 1 | 5.3,[0.9 - 24.6] | 15 | 0 | 0 |

**Table 2.** As in Table 1 but for drifters during the spawning season (September-May) and for 3 ranges of PPLD (results shown in Fig. 4). Results are grouped for PPLD ranges, showing for each range: the number (N) of drifters that entered or were launched in a given SA, the number of drifters that continued from the SA to the Gulf of Manfredonia (N. Gulf), and the percentage (%) of N Gulf drifters with respect to N, together with the 95% confidence intervals estimated using the Wilson Score.

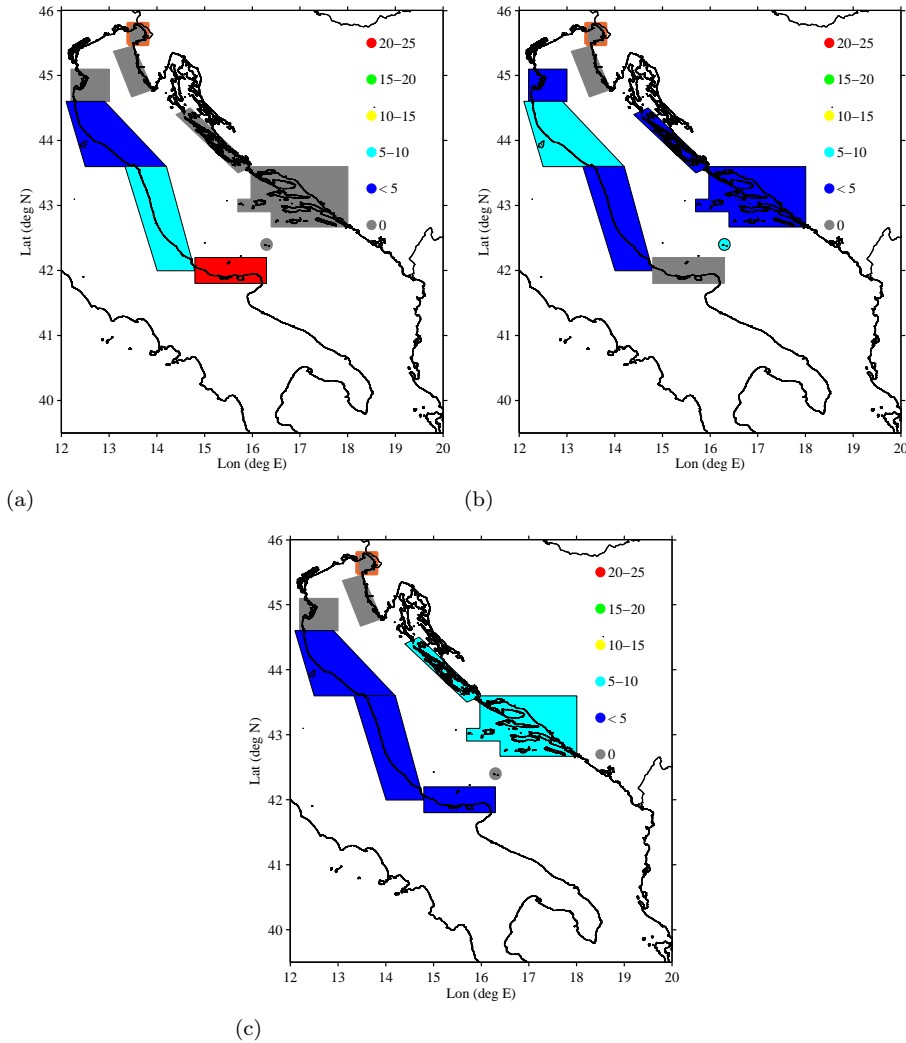

**Figure 7.** The connection percentages (i.e. percentage of drifters that passed through a given SA and reached the Gulf of Manfredonia) for sardine spawning season (see Table 2) over the following PPLD ranges: (a) 0 - 20 days; (b) 20 - 40 days; (c) 40 - 60 days. Results obtained with less than 10 drifters per SA, as for the Gulf of Trieste (SA 4, see Table 2) , are disregarded and identified by the orange contour around the SA.

1-3) as well as Palagruža Islands (SA 9) show no connection. This overall pattern is significant despite considering a drifter subset based on a specific PPLD range. In fact, drifters have to follow the central recirculating sub-basin gyre before reaching the Gulf which requires a travel time longer than 20 days.

As the PPLD range increases to 20-40 days (Fig. 4b), the Gargano Promontory SA shows zero connection, while the other Western SAs as well as the Eastern SAs except for the Istrian Peninsula (SA 3) show significant connections. Given the wide 95% confidence intervals, the details of the distribution cannot be trusted but the differences between SA 8 and the other SAs 6, 9 and 1 are significant, indicating that the general pattern can be trusted. The lack of connection with the Gargano Promontory is due to the fact that particles reach the Gulf too quickly with respect to the PPLD. Drifters coming from both the Western and Eastern coast are carried out by the central sub-basin gyre.

Finally, for the highest PPLD range 40-60 days (Fig. 4c), the Gargano Promontory SA shows some connection, similarly to the other Western and Eastern SAs influenced by the central sub-basin gyre. These connections are significantly different from zero, even though the details of the differences between the Eastern and Western values are not significant. The result suggests that drifters at these PPLDs reach the Gulf through the recirculation, even when coming from Gargano Promontory, possibly looping more than once. The zero connection from the Istrian Peninsula confirms the results of the other PPLDs, suggesting that the area is poorly connected to the Gulf. Also Po River Delta (SA 5) and Palagruža Islands (SA 9) have consistently low or zero connection values.

In summary, the results show a marked dependence on the PPLD. For short PPLD the Gargano Promontory SA is the most likely contributor to the Gulf of Manfredonia, followed by the other Western SAs. At increasing (and more realistic) PPLDs the SAs along the central sub-basin gyre are the best contributors, both along the Western and Eastern coast. The Northern SAs, and especially the Istrian Peninsula, are poorly connected for all PPLDs. The Palagruža Islands appears to be only marginally connected, despite their proximity to the Gulf.

## 4.3 Retention properties of the Gulf of Manfredonia

### 4.3.1 Drifter analysis

A first analysis of the retention properties of the Gulf is performed using drifter data. A total of 32 drifters from the complete dataset entered or were launched in the Gulf. Twenty-six of these drifters were present in the Gulf during the sardine spawning period. The distribution of their data density in the Gulf (Fig. 8a) provides a first indication of the most likely areas sampled by the drifters. Maximum drifter concentrations are found close to the tip of the Gargano promontory, suggesting that most drifters enter the Gulf following the WAC. This is shown also by the individual drifter trajectories in Fig. 8b (colour coded as function of time before and after entering the Gulf). Exit pathways are more distributed in space, with some of the drifters exiting Southward along the coast while others move offshore, Eastward and sometime Northward from the Gulf. These different pathways are likely to be related to different wind conditions (Specchiulli et al., 2016). Northwesterly (downwelling prone) winds tend to reinforce the boundary current flowing Southward along the coast, while Southeasterly (upwelling prone) winds

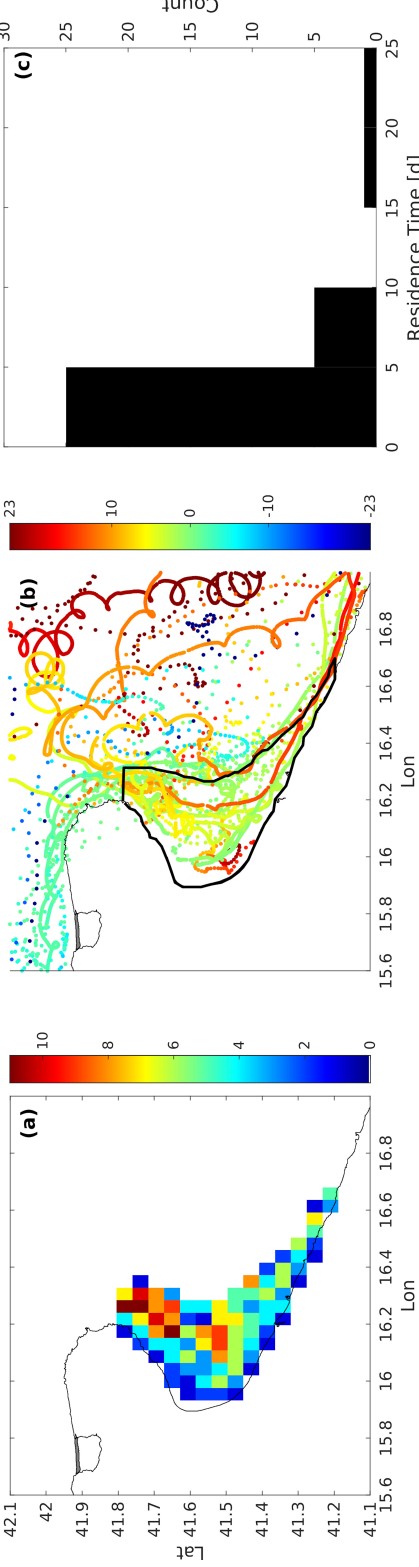

**Figure 8.** (a) Density of drifter data in the Gulf of Manfredonia, computed as the number of nonconsecutive entries into each 0.044° bin. (b) Trajectories of drifters that entered the Gulf of Manfredonia color coded with respect to time of first entry in the Gulf. The color scale corresponds to days before/after the first entry in the Gulf of Manfredonia. The boundaries of the Gulf, as defined in Section 3.4, are represented by the thick black line. (c) Histogram of drifter residence times in the Gulf of Manfredonia

| Start | End | $N_R$ | N Trajectories |
|-------|-----|-------|----------------|
| 18/10/13 | 31/10/13 | 28 | 11032 |
| 1/11/13 | 30/11/13 | 60 | 23640 |
| 1/12/13 | 31/12/13 | 62 | 24428 |
| 1/01/14 | 31/01/14 | 62 | 24428 |
| 1/02/14 | 28/02/14 | 56 | 22064 |
| 1/03/14 | 31/03/14 | 62 | 24428 |
| 1/04/14 | 30/04/14 | 60 | 23640 |
| 1/05/14 | 31/05/14 | 62 | 24428 |
| 1/06/14 | 30/06/14 | 60 | 23640 |
| 1/07/14 | 10/07/14 | 20 | 7880 |

**Table 3.** The start and end dates of the particle tracking period used to estimate monthly residence times in the Gulf of Manfredonia. Particles were released at 394 initial positions in the Gulf of Manfredonia, between the start and end dates, re-released every 12 hours and tracked for 30 days. $N_R$ denotes the number of releases and N Trajectories the total number of trajectories for each month defined as $N_R \times 394$. Note that in July particles were tracked for 20 days due to large gaps in the HF radar velocity data.

tend to induce offshore transport disrupting the boundary current and possibly even reverse it (see Figure 7 of Carlson et al. 2016).

The histogram of residence times for these drifters is shown in Fig. 8c. 25 of the 32 drifters have residence times up to 5 days, 5 between 5 and 10 days, and 2 exceeded two weeks. The mean residence time in the Gulf is 3.5 days. These statistics
should be considered with some caution and taken as a qualitative indication, since they are based on a relatively small data set and also they are biased for two different reasons. Six of the 32 drifters stopped functioning in the Gulf, presumably as a result of grounding, battery failure, and/or interference from fishermen and other seafarers, therefore inducing a bias toward small values. On the other hand, five of the drifters were launched within the Gulf specifically targeting retention areas (Corgnati et al., 2018), therefore inducing a bias toward high values.

**4.3.2 Residence times from HF radar**

A more robust estimate of residence times is computed using virtual particles advected in the HF radar velocity field. Details on the launching dates are provided in Table 3. Results are shown in Fig. 9 and 10 in terms of monthly means and standard deviation (*std*), computed in each cell of the radar grid using all the particles launched from that cell.

For each month, the highest residence times (Fig. 9) are found in the central area of the Gulf. Residence times are typi-
20 cally less than 5-6 days, except during the month of October when they exceed 10 days. These results are in line with the observed surface currents in the Gulf of Manfredonia. In fact, months with high (October)/low (February) residence times show weaker/stronger surface currents in the central area of the Gulf (Fig.2).

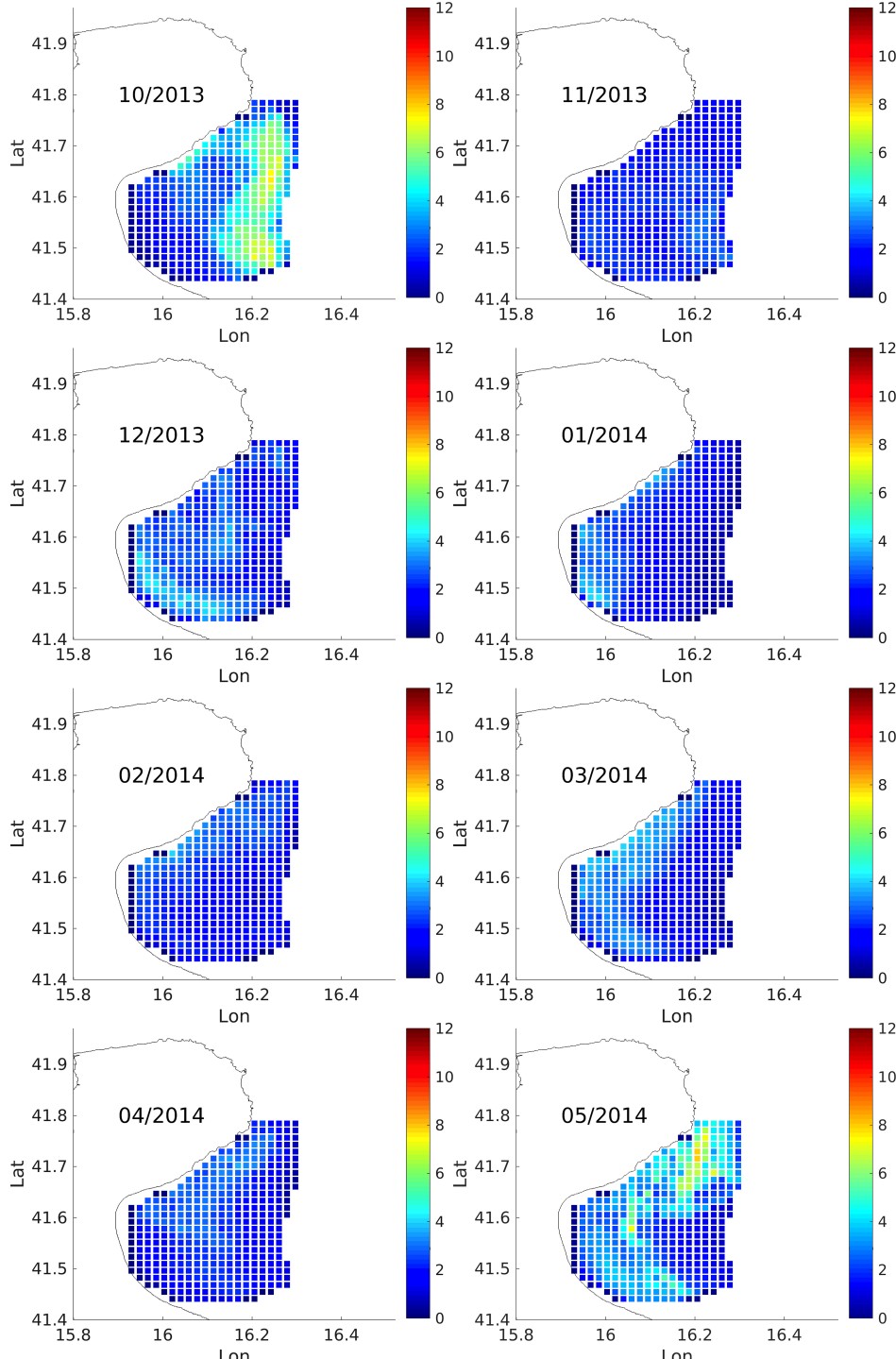

**Figure 9.** Monthly bootstrap estimates of average residence times (in days) of virtual particles advected in the HF radar velocity field and released within the boundaries of the Gulf of Manfredonia. The date ranges, number of releases, and total number of particles tracked per month are summarized in Table 3.

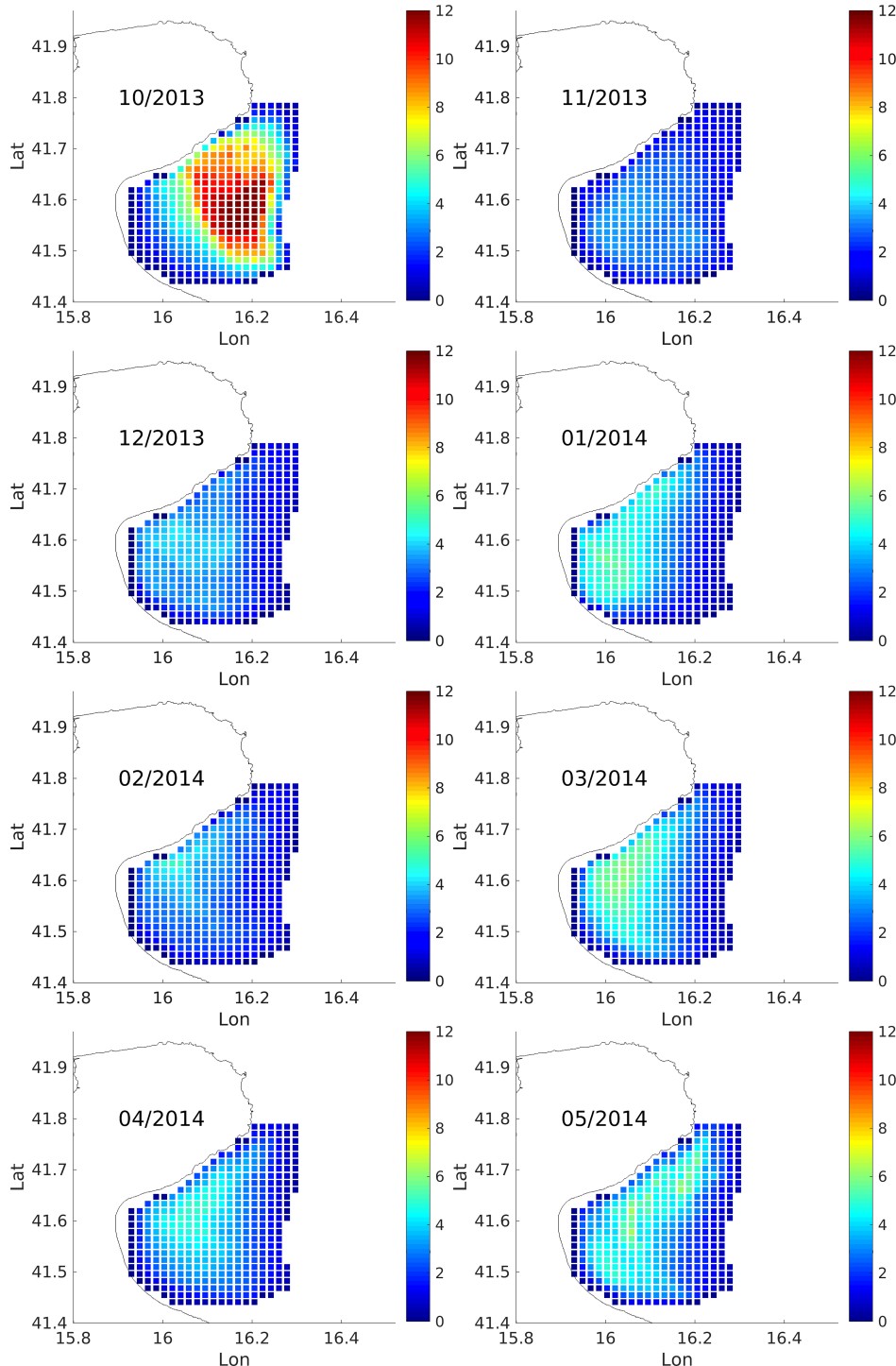

**Figure 10.** As for Fig. 8 but for residence time standard deviations (in days).

The *stds* (Fig. 10) are typically below 4 days, reaching values of 6-7 days only in October and in May. All together, these values are qualitatively compatible with those shown by the drifters (Fig. 8). These residence times as well as those computed from surface drifters are robust with respect to the definition of Gulf of Manfredonia.We shifted the eastern boundary offshore (i.e. 30 m isobath) and the results are qualitative the same.

The results show that particles passively advected by surface currents typically spend less than 10 days in the Gulf before they exit. This in turns suggests that, given the expected PPLD values for sardines and given the age of the larvae observed in the Gulf, it is unlikely that the nursery area is primarily supported by local spawning. The smaller individuals (i.e. 15-22 mm TL, Fig. 4) sampled inside the Gulf were about 20-40 days old, consequently, external SAs connected with the Gulf through advection pathways are likely to play a fundamental role in maintaining the nursery.

It is worth mentioning that recently Bray et al. (2017) found high retention values within the Gulf of Manfredonia, up to 20 days. However, these findings are based on a relatively coarse resolution model, $1/16°$, which equates to approximately 5.25 km. The model, therefore, may not resolve the dynamics of the Gulf. By comparison, the 1.5 km spatial resolution of the HF radar-derived velocities is 3.5 times higher, and likely resolves the relevant dynamics.

## 4.4 Discussion on surface and water column

The results presented here are based on drifters and HF radars that sample surface velocity in the first meter of water. Larvae of sardines, on the other hand, are known to migrate vertically in the upper 15-20 m of the water column, mostly concentrating at surface ($\sim$ 5-10 m) during the night and dispersing through the water column during the day (Olivar et al., 2001; Santos et al., 2006). Therefore, an important question may arise: how representative are surface drifters and HF radar data of the upper 15-20 m of the water column? Here we provide a first qualitative assessment of this issue, considering literature results on current profiles taken from ADCPs in the Adriatic in the upper 15-20 m. The spatial distribution of available ADCP results is quite sparse, and the measurements typically do not reach the surface (in order to avoid side lobe contamination), so the following comparison is necessarily qualitative.

In winter, typical mixed layer depths in the Adriatic are between 50 m and 100 m everywhere except for the northern Adriatic where typical values are around 30 m (Artegiani et al., 1997a; D'Ortenzio et al., 2005) Pullen et al. (2003) and Book et al. (2005) show results from ADCP measurements taken in the North/central WAC during winter 2001 (January- April) at a depth of approximately 60 m. The instrument setting provides an upper limit of $\sim$ 6 m below the sea surface, and the results show that the current is primarily barotropic, at least on average, over the water column.

Grilli et al. (2013) report results from an extensive array of 12 ADCPs deployed in the framework of the DART (Dynamic of the Adriatic in Real Time) experiment along a transect from the Gargano to the Northern Dalmatian Islands, maintained from October 2005 to November 2006. The upper depth limit of the measurements varies depending on settings and on the depth of each site, and it ranges between 2 m and 15 m. Results in the period November-April indicate that in most sites the water column is primarily barotropic in the upper 20 m, even though some vertical shear is evident especially for the most coastal sites, with velocity gradients up to approximately 5 cm/sec in $\sim$ 20 m. The ADCP measurements are in line with the general

picture suggested by Artegiani et al. (1997b) for which temperature and salinity compensation effects, give rise to a winter dynamics in the Adriatic dominated by a barotropic, wind-induced transport and circulation.

A detailed study has been carried out by Corgnati et al. (2018) in the Gulf of Manfredonia, comparing results from HF radar and ADCP. The ADCP was deployed in November 2014 in a central area of the Gulf at a depth of 17 m. It was set to provide velocity information below 1 m from the surface, with a resolution of 5 m resulting in 3 depth cells between 1 and 16 m. The comparison is limited to 2 periods of 10 days each, during January and March 2015, because after November 2014 the radar was discontinuously operated due to problems in two of the sites. The results show that the water column is mostly barotropic,

with correlation coefficients between the 3 cells greater than 0.90, even though one baroclinic episode is observed during the January period. Comparison with the HF radar velocity at the grid cell corresponding to the ADCP position shows a good correlation (in the range of 0.76 - 0.95). The HF radar velocities are typically higher than the velocities in the water column, as it can be expected given that the surface is in direct contact with atmospheric forcing, and the attenuation coefficients between surface and bottom cell can reach $\sim 30\%$.

In summary, the results indicate that at least during the winter period the velocity field is primarily barotropic in the upper 15-20 m with a good correlation with the surface velocity, even though sporadic baroclinic events can occur (Specchiulli et al., 2016). This suggests that the surface results obtained here are relevant also for the entire upper water column, especially in

terms of mean pathways and connections. Vertical shear is likely to occur close to the surface that is directly influenced by the atmosphere. As a consequence, residence time values could be underestimated with respect to those obtained with vertical migration, even though the differences are not expected to alter the overall conclusions.

## 5    Conclusions

In this work, the processes of surface advection that contribute to maintain the sardine nursery in the Gulf of Manfredonia are

investigated. Two surface velocity data sets, provided by the Adriatic historical drifters and by a HF radar system in the Gulf, are considered during the sardine spawning period (September-May). Drifter data are used to study the transport from remote spawning areas (SAs) in the central and Northern Adriatic toward the Gulf. Results show a strong dependence on the PPLD parameter. Since growth rates may differ at different temporal and spatial scales in response to changes in environmental factors (such as temperature and food availability), and since the influence of transport depends on temporal scales, three extended

PPLD ranges are considered. For short PPLDs (less than 20 days), the highest connection with the Gulf is found for the Gargano Promontory SA, just North of the Gulf. The other SAs along the Western coast are also connected, with values decreasing in the Northward direction. This indicates that eggs and larvae are carried out mostly through the Southward boundary current WAC. At increasing and more realistic PPLDs (ranges 20-40, 40-60 days), the SAs situated on the central Eastern coast are also connected with the Gulf with values similar to the Western ones, indicating that eggs and larvae are likely to be advected

through the central recirculating sub-basin gyre. The northernmost SAs, (and in particular the Istrian Peninsula) are poorly or not at all connected, suggesting that the Northern sub-basin gyre is isolated or that the associated transit times are longer than the considered PPLDs. Also the Palagruža Islands SA has an overall low connection with the Gulf, even though it is

one of the closest SAs from a geographical point of view. This is an example of the difference between geographic distance and 'oceanographic distance' (Jönsson and Watson, 2016), that is related to the transport time due to the current and whose importance has been pointed out in several biological applications. The Palagruža Islands SA is indeed one of the farthest SAs from the oceanographic point of view, likely because it is only occasionally involved in the Eastward limb of the central sub-basin gyre, that has high variability and reduced transport with respect to the most prominent Westward arm (Carlson et al., 2016). Indeed, the Palagruža Islands are located close to a known hyperbolic point (Veneziani et al., 2007; Haza et al., 2007) in the Adriatic Sea circulation, that is characterised by high variability and transport uncertainty.

HF radar data were used to compute residence times in the Gulf, using virtual particles launched and advected in the radar velocity fields. Results show that average residence times in the Gulf are typically less than $\sim$ 5-6 days, with *std* less than $\sim$ 4 days. This is in agreement with the finding of Veneziani et al. (2007), showing that drifters typically spend a maximum of $\sim$10 days in the enlarged area that encloses the Gulf and the Gargano. Only during the month of October, average residence times are found to exceed 10 days, while *std* reaches 6-7 days. Overall, the results indicate that particles are typically trapped in the Gulf for periods less than 10 days, i.e. shorter than the typical sardine PPLDs. It is interesting to compare the residence time values in Manfredonia with the advection time scale $T_{adv} = L/U$ (where $L$ and $U$ are typical length and velocity scales). If we consider a space scale $L$ comparable with the size of the Manfredonia Gulf, $L \sim$ 30-40 km and a typical average velocity of the WAC $U \sim 20 - 30$ cm/s (Veneziani et al., 2007; Poulain, 2001), we obtain $T_{adv} \sim$ 1-2 days , which is significantly shorter than the estimated Gulf residence times. This difference helps explaining why the Gulf is characterised by highest larvae concentration with respect to the other coastal regions (Borme et al., 2013). Compared to the WAC, the recirculation in the Gulf of Manfredonia acts as a retention point.

In summary, the results presented here and obtained in the framework of the JERICO-NEXT project, have shown that HF radar data can be used to asses biological transport and have relevant implications for the study of fish recruitment in costal areas and more specifically for the sardine nursery of the Gulf of Manfredonia. The residence times shorter than the typical PPLDs suggest that is unlikely that the nursery is mostly supported by local spawning, while larvae are likely to be advected from remote SAs. The SAs that mostly contribute to the nursery are likely to be the ones situated along the central sub-basin gyre, on both the western and Eastern coasts of the Adriatic. The northernmost SAs and the Palagruža Islands are less connected and less likely to contribute to the maintaing of the nursery.

This supports the idea of the evolutionary selection of an opportunistic reproductive strategy for the sardine species, consisting of high fecundity associated with an extended spawning period over a wide area. This strategy allows a continual 'testing' of favourable environmental conditions and increases the chances that at least some eggs or larvae are present when environmental conditions are favourable for survival. In addition, concentration, retention and enrichment are considered the 'triad' of oceanographic conditions and can largely contribute to recruitment variability (Bakun, 1996). Both enrichment (upwelling, mixing, etc.) and concentration processes (convergence, fronts, etc.) lead adult individuals to areas where there is enough food for larvae, whilst dispersal/retention of early-life stages, which mainly depend on circulation patterns and transport pathways from the spawning grounds to the nursery areas, can largely enable the larvae to stay in these favourable areas. In order to

ensure successful transport, therefore, there must be some synchronization between physical process (current circulation) and biological process (i.e. spawning grounds and period).

While the datasets employed here are the most comprehensive available for the Adriatic Sea and the Gulf of Manfredonia, they are, of course, subject to limitations. Most importantly, both drifter and HF radar data measure currents in approximately the first meter of water while sardine ELHS are known to exhibit diel migrations in the upper 15-20 m of the water column. However, the literature survey of ADCP data indicates that at least during the winter period the flow is mostly barotropic with a good correlation in the first 20 m, indicating that our results are relevant also for the upper water column. A second, and related, limitation of the present results is that they focus on the physical advection processes only while no biological behaviour, such as active swimming, is included. However, a recent study by Silva et al. (2014) evaluated larvae swimming speed. Swimming performance of sardine larvae increased significantly with the ontogeny, reaching a maximum of 9.47 cm s$^{-1}$ approximately 55 days after hatching. This value is significantly smaller than the typical average velocity of the WAC $\sim 20$ - 30 cm s$^{-1}$ (Veneziani et al., 2007; Poulain and Cushman-Roisin, 2001), confirming that ELHS are likly to be transported by ocean currents. The PPLD is the only biological parameter considered, as a basis for the advection times used in the analysis. Lastly, the drifter analysis presented here is based on simple direct connectivity metrics. Recently, surface drifters trajectories have been analyzed using more sophisticated methods such as the multi-iteration and transit time methods (Maximenko et al., 2012; Rypina et al., 2017) and applied to the spreading of pollutants (van Sebille et al., 2012, 2015; Rypina et al., 2017). The applicability of these methods to a biological context is deferred to future work. Despite these limitations, our results are the first dedicated effort based on oceanographic data to investigate advection properties relevant to the Gulf of Manfredonia nursery and provide a data-based benchmark that can be used, for example, to validate future modeling efforts that attempt to simulate ELHS behaviour.

*Data availability.* The datasets used in this study can be obtained at : http://doga.ogs.trieste.it/sire/medsvp/ (Drifters), http://radarhf.ismar. cnr.it (HF Radar).

*Competing interests.* No competing interests are present

*Acknowledgements.* This project has been supported and co-financed by the JERICO-NEXT project. This project has received funding from the European Union's Horizon 2020 research and innovation program under grant agreement no. 654410. The experiments have been carried out within the COCONET project (Grant agreement No. 287844), and the Italian national projects SSD-PESCA and flagship project RITMARE.

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
