# Peer review of "Linking sardine recruitment in coastal areas to ocean currents using surface drifters and HF radar. A case study in the Gulf of Manfredonia, Adriatic Sea."

_Ocean Science, 2018_

## Referee Comment (RC1) · Anonymous Referee #1 · 26 Sep 2018

Review of the manuscript "Linking sardine recruitment in coastal areas to ocean currents using surface drifters and HF radar. A case study in the Gulf of Manfredonia, Adriatic Sea" by Sciascia et al

The paper investigates the residence times of passively-advected particles within the Gulf of Manfredonia (GoM) and the connectivity of the GoM to other coastal areas of the Adriatic using CODE surface drifters and HF radar. Assuming that sardine eggs/larvae are passive particles advected by oceanic surface currents, the results can be interpreted as the retention times of larvae inside the GoM and import of larvae from other

spawning areas (SA's). Short retention times and relatively large connectivity values resulting from this analysis suggest that larvae in the GoM are not likely to be spawned there and are more likely to get advected from other SA's.

The paper is interesting, easy to read, and presents clear conclusions. However, I recommend addressing several comments and questions below before it can be published.

1) Connectivity analysis is based only on "direct" drifter trajectory segments connecting 9 SA's to GoM. It would be interesting to see if slightly more sophisticated mathematical methods such as, for example, the transit matrix approach (where the probability matrix showing connectivity between different bins over short time T_transit is constructed based on drifters, and then used iteratively to obtain connectivity over longer times) or multi-iteration approach (where segments of different trajectories that run through the same bin are stitched together to construct a more comprehensive drifter dataset) would give similar results.

van Sebille, E., M. H. England, and G. Froyland, 2012: Origin, dynamics and evolution of ocean garbage patches from observed surface drifters. Environ. Res. Lett., 7, 044040, doi:10.1088/1748-9326/7/4/044040.

van Sebille, E., and Coauthors, 2015: A global inventory of small floating plastic debris. Environ. Res. Lett., 10, 124006, doi:10.1088/ 1748-9326/10/12/124006.

Maximenko, N. A., J. Hafner, and P. P. Niiler, 2012: Pathways of marine debris derived from trajectories of Lagrangian drifters. Mar. Pollut. Bull., 65, 51–62, doi:10.1016/j.marpolbul.2011.04.016.

Rypina, I. I., D. Fertitta, A. Macdonald, S. Yoshida, S. Jayne (2016). Multi-iteration approach to studying tracer spreading using drifter data. J. Phys. Oceangr., 47, 339–351,doi: 10.1175/JPO-D-16-0165.1.

2) What is the average lifetime of the drifters? And how does it compare to the transit

times from different SA's to GoM? If comparable, could it influence/bias the results? The two methods mentioned above could help eliminating/reducing this possible bias.

3) It would be helpful to show mean+/-std transit times from the 9 SA's to GoM in Fig. 5c.

4) It is not clear whether absence of eggs and larvae <15 mm in the GoM during the 2 cruises indicates that sardines do not spawn there, or that sardines do spawn there but the eggs and larvae have been advected out of GoM by the time of the cruises.

Have eggs ever been found in the GoM?

Is GoM a known spawning area for sardines?

5) Would other factors such as water temperature, salinity and nutrients be important for spawning in the GoM and for the larval development, and if so, do conditions in the GoM meet the necessary requirements?

6) What would be the role of active horizontal swimming of larvae? Is active swimming speed of larvae comparable or much smaller than the typical velocities in the GoM and other areas of the Adriatic?

7) Authors suggest that larval vertical migration down to 20 m will not change their results because the currents in the GoM and in many parts of the Adriatic are primarily barotropic and thus depth independent.

What is the mixed layer depth in winter throughout the Adriatic? If <20m, why would the currents be the same within and below the mixed layer (since physics is different for the mixed layer vs below)?

Please provide an explanation about why the currents would be primarily depth-independent down to 20 m throughout most of the Adriatic.

8) Since the paper focuses on GoM, more explanation/info about the physical oceanography of the GoM would be useful – its stratification, major forcing mechanisms, influence of local vs remote forcing on the circulation, variability of the currents throughout the year, occurrence of eddies and recirculations, importance of tides etc.

9) A picture of the mean circulation in the GoM based on HF radar data would be useful, perhaps with 4 panels showing the mean currents during the 4 seasons.

10) Have deeper SVP drifters been deployed in the Adriatic? How do trajectories of CODE drifters compare to SVP drifters?

11) Radar resolution of 1.5 km might be too coarse to fully resolve near-shore processes. Perhaps, a comparison between real and HF-simulated drifters (ensemble-averaged separation between real-simulated drifters as a function of time, or something similar) could be shown to investigate how well the radar represents the actual currents?

12) The 25 m isobath seems a little arbitrary as the outer boundary of the GoM. Would the residence times change a lot if, say, a 30-m isobath is used instead? Same question about the northern and southern boundaries of the bay – would results change drastically if these boundaries are moved a bit?

---

## Referee Comment (RC2) · Anonymous Referee #2 · 27 Sep 2018

The manuscript addresses the influence of sea currents on recruitment in the Gulf of Manfredonia using information based on current measurement data sampled the the first meter of water. Authors investigate on recruitment dynamics in the Gulf

---

## Editor Comment (EC1) · I. Puillat (Editor) · 9 Oct 2018

Dear Reviewer

we are very grateful for your review but it seems your report have been truncated after a few lines. Would it be possible to send it again please? You can also use my email address if you find it easier: ingrid.puillat@ifremer.fr

Thanks in advance for your support

[Figure]

Best regards

ingrid puillat

---

## Author Comment (AC1) · 24 Oct 2018

*Dear Editor,*

*Enclosed herewith please find the revision of the manuscript entitled: "Linking sardine recruitment in coastal areas to ocean currents using surface drifters and HF radar. A case study in the Gulf of Manfredonia, Adriatic Sea." , referenced as os-2018-65. We thank the reviewers for their thorough reading of the text and comments. Below is our detailed response (in* **bold***) while modifications to the manuscript are highlighted in* **blue** *both in the new version and in this response. Unfortunately we haven't received the complete review from Reviewer #2 however, the title of the review states that the manuscript only needed misprint correction. In the revised version of the manuscript we corrected typos, modified the text to clarify the points raised by Reviewer #1 and added a new figure, now Figure 2, and modified the old Figure 5c,d now Figure 6c,d.*

*We hope that the manuscript is now satisfactory for publication in Ocean Science.*

*Best Regards,*

*On behalf of all authors*
*Roberta Sciascia*

**Anonymous Referee #1**

Review of the manuscript "Linking sardine recruitment in coastal areas to ocean currents using surface drifters and HF radar. A case study in the Gulf of Manfredonia, Adriatic Sea" by Sciascia et al

The paper investigates the residence times of passively-advected particles within the Gulf of Manfredonia (GoM) and the connectivity of the GoM to other coastal areas of the Adriatic using CODE surface drifters and HF radar. Assuming that sardine eggs/larvae are passive particles advected by oceanic surface currents, the results can be interpreted as the retention times of larvae inside the GoM and import of larvae from other spawning areas (SA's). Short retention times and relatively large connectivity values resulting from this analysis suggest that larvae in the GoM are not likely to be spawned there and are more likely to get advected from other SA's.
The paper is interesting, easy to read, and presents clear conclusions. However, I recommend addressing several comments and questions below before it can be published.

1) Connectivity analysis is based only on "direct" drifter trajectory segments connecting 9 SA's to GoM. It would be interesting to see if slightly more sophisticated mathematical methods such as, for example, the transit matrix approach (where the probability matrix showing connectivity between different bins over short time T_transit is constructed based on drifters, and then used iteratively to obtain connectivity over longer times) or multi-iteration approach (where segments of different trajectories that run through the same bin are stitched together to construct a more comprehensive drifter dataset) would give similar results.

van Sebille, E., M. H. England, and G. Froyland, 2012: Origin, dynamics and evolution of ocean garbage patches from observed surface drifters. Environ. Res. Lett., 7, 044040, doi:10.1088/1748-9326/7/4/044040.

van Sebille, E., and Coauthors, 2015: A global inventory of small floating plastic debris. Environ. Res. Lett., 10, 124006, doi:10.1088/ 1748-9326/10/12/124006.

Maximenko, N. A., J. Hafner, and P. P. Niiler, 2012: Pathways of marine debris derived from trajectories of Lagrangian drifters. Mar. Pollut. Bull., 65, 51–62, doi:10.1016/j.marpolbul.2011.04.016.

Rypina, I. I., D. Fertitta, A. Macdonald, S. Yoshida, S. Jayne (2016). Multi-iteration approach to studying tracer spreading using drifter data. J. Phys. Oceangr., 47, 339– 351,doi: 10.1175/JPO-D-16-0165.1.

**R: Yes, in this manuscript we only considered the direct connectivity as we are focusing on trajectories leaving specific coastal areas and reaching a localized target, instead of looking at the spreading of particles over the whole basin.**

**We thank the reviewer for the important comment on the more advanced methods, and we have inserted it together with the suggested references in the revised version. Applications of the advanced methods will be consider in future analysis. (pg. 27, l. 12-15) "Lastly, the drifter analysis presented here is based on a simple direct connectivity metrics. Recently, surface drifters trajectories have been analyzed using more sophisticated methods such as the multi-iteration and transit time methods (Maximenko et al, 2012; Rypina et al., 2016) and applied to the spreading of pollutants (van Sebille et al 2012; 2015; Rypina et al., 2016). The applicability of these methods to a biological context is deferred to future work."**

2) What is the average lifetime of the drifters? And how does it compare to the transit times from different SA's to GoM? If comparable, could it influence/bias the results? The two methods mentioned above could help eliminating/reducing this possible bias.

**R: The transit times are not affected by the drifters lifetime and to address this comment we have modified panel d of Figure 5 that now shows the transit times of the drifters that reached the Gulf of Manfredonia. The average lifetime of the drifters entering the GoM is 90 days but transit times longer than 60 days are not biologically relevant as the larvae would, at that point, have transitioned to an active swimming phase. In this version of the manuscript we added the following sentences (pg. 12, l. 14-15) "Note that the average lifetime (~90 days) of drifters entering the Gulf of Manfredonia is well above the maximum PPLD considered in the analysis." and (pg. 16, l. 16-23)**
**"In order to gain some insight on the time scales involved, Fig.5c also indicates the average time required to reach the Gulf from each SA. Drifters leaving SA 8 need on average 13 days, while the SAs located on the western Adriatic coast need on average 35 days and SA located on the eastern coast need about 50 days. Even though average times for some SAs are not quantitatively significant given the fractional number of drifters reaching the Gulf (see Table 1), results are consistent with the unconditioned statistics (Fig. 5d) showing the average transit time of drifters reaching the Gulf from the whole Adriatic Sea. Most of the drifters coming from the area North of the Gulf need 10 days or less in agreement with the WAC southward flow. On the other hand, drifters coming from the Northern-Central basin need longer time. This is a consequence of the advection from the recirculating sub-basin gyres".**

3) It would be helpful to show mean+/-std transit times from the 9 SA's to GoM in Fig. 5c.

**R: In line with comment above in the new version of the manuscript we modified Figure 5c and estimated mean transit times to the GoM from each SA. Notice that the computation of average transit times is an important general indication of the involved scales, as shown also by the consistency between Fig.5c and 5d. On the other hand, the average transit times in Fig.5c cannot be considered as quantitatively significant and cannot be characterized using for instance std, because of the small number of drifters reaching the GoM as shown in Table 1.**

4) It is not clear whether absence of eggs and larvae <15 mm in the GoM during the 2 cruises indicates that sardines do not spawn there, or that sardines do spawn there but the eggs and larvae have been advected out of GoM by the time of the cruises.
Have eggs ever been found in the GoM?
Is GoM a known spawning area for sardines?
**R: The GoM has been typically described in the literature as a nursery area fostering larvae and juveniles which were spawned elsewhere and advected to the GoM through the main currents (Morello and Arneri, 2009; Borme et al., 2013). In particular, previous systematic ichthyoplanktonic surveys showed great abundance of sardine post-larvae and juveniles, whereas the number of eggs found in the GoM was low.**
**Nevertheless, a complete study addressing the question of the dynamics of the Gulf and the impact of larval advection from and to the Gulf were still lacking and motivate the present paper.**
**In this version of the manuscript we addressed this comment together with point 5 by adding the following sentence (pg. 7, l. 20-22): "The Gulf of Manfredonia has been typically described as a nursery area fostering larvae and juveniles spawned elsewhere (Morello and Arneri, 2009). Ichthyoplanktonic surveys showed great abundance of sardine post-larvae and juveniles, whereas the number of eggs found in the Gulf was low (Panfili, 2012; Borme et al., 2013). "**

5) Would other factors such as water temperature, salinity and nutrients be important for spawning in the GoM and for the larval development, and if so, do conditions in the GoM meet the necessary requirements?
**R: Yes, the reviewer is right and as we mentioned in section 2.3 water temperature, salinity and nutrients are certainly the most important parameters in determining spawning grounds (Morello and Arneri, 2009).**
**Within the GoM these parameters might not be suitable during the winter reproductive months in the Gulf, while they are suitable for nursery during the following months.**
**In particular, river discharge and rains tend to freshen the waters inside the GoM thus resulting in less favorable conditions compared to other areas of the Adriatic (Morello and Arnieri, 2009).**
**In this version of the manuscript we clarified this important point (pg. 7, l. 22-25). " In fact, environmental conditions in the Gulf of Manfredonia might not be suitable during the winter reproductive months, while they are suitable for nursery during the following months. In particular due to river discharge and rains, the waters inside the Gulf are fresher than the waters in other areas of the Adriatic (Morello and Arneri, 2009)."**

6) What would be the role of active horizontal swimming of larvae? Is active swimming speed of larvae comparable or much smaller than the typical velocities in the GoM and other areas of the Adriatic?

**R: A recent study on swimming behaviour in sardine larvae was carried out by Silva et al., (2014). Critical swimming speed was evaluated by rearing larvae under different feeding conditions, optimal temperature (15 °C) and salinity (35). Swimming performance of sardine larvae increased significantly with the ontogeny, reaching a maximum of 9.47 cm s$^{-1}$ at 19.10 mm TL and 55 days after hatching. This value, which corresponds to the latest stages of the larvae life, is significantly smaller than the typical average velocity of WAC ~ 20 – 30 cm/s (Veneziani et al., 2007; Poulain, 2001), confirming that the early life stages are "necessarily" transported by the main currents flow.**
**In this version of the manuscript we added the following sentence (pg. 27, l. 7-11): " However, a recent study by Silva et al., 2014 evaluated larvae swimming speed. Swimming performance of sardine larvae increased significantly with the ontogeny, reaching a maximum of 9.47 cm s$^{-1}$ approximately 55 days after hatching. This value is significantly smaller than the typical average velocity of the WAC ~ 20 – 30 cm/s (Veneziani et al., 2007; Poulain, 2001), confirming that ELHS must be transported by the ocean currents."**

7) Authors suggest that larval vertical migration down to 20 m will not change their results because the currents in the GoM and in many parts of the Adriatic are primarily barotropic and thus depth independent.
What is the mixed layer depth in winter throughout the Adriatic? If <20m, why would the currents be the same within and below the mixed layer (since physics is different for the mixed layer vs below)? Please provide an explanation about why the currents would be primarily depth- independent down to 20 m throughout most of the Adriatic.
**R: We thank the reviewer for the question that helps us elucidate an important point of the manuscript.**
**In winter, typical mixed layer depth in the Adriatic are between 50m and 100m everywhere except for the northern Adriatic where typical values are around 30m (Artegiani et al., 1997a, D'Ortenzio et al., 2005). Artegiani et al., 1997b suggested that the general circulation is dominated by temperature and salinity compensation effects, which give no resulting density signal and speculate that during winter the barotropic, wind-induced transport and circulation is probably a major component of the general circulation.**
**In this version of the manuscript we clarified this key point adding the following sentences (pg. 24, l. 23-24) "In winter, typical mixed layer depth in the Adriatic are between 50m and 100m everywhere except for the northern Adriatic where typical values are around 30m (Artegiani et al., 1997a, D'Ortenzio et al., 2005). " and (pg. 24, l. 33, pg. 25, l. 1-2) "The ADCP measurements are in line with the general picture suggested by Artegiani et al., 1997b for which temperature and salinity compensation effects, give rise to a winter dynamics in the Adriatic dominated by a barotropic, wind-induced transport and circulation."**

8) Since the paper focuses on GoM, more explanation/info about the physical oceanography of the GoM would be useful – its stratification, major forcing mechanisms, influence of local vs remote forcing on the circulation, variability of the currents throughout the year, occurrence of eddies and recirculations, importance of tides etc.

**R: To our knowledge section 2.2 of the manuscript is a comprehensive review of the available literature on the Gulf of Manfredonia. The Gulf is a relatively understudied area and the available peer-reviewed hydrographic data, circulation patterns and forcings are summarized in this section. Previous studies like Burrage et al., (2009) largely focused on the instabilities of the buoyant boundary current outside the Gulf and not specifically on the dynamics within the Gulf.**
**In this version of the manuscript we have added a more detailed description of the water properties (temperature, salinity) throughout the year: (pg. 5, l. 21-25)"In winter, temperature in the Gulf is approximately 10°C (Broome et al., 2013; Casale et al., 2012). Surface temperatures vary from 14-18.5°C in April - May (Campanelli et al., 2013; Casale et al., 2012; Monticelli et al., 2014), increase to over 23°C in June (Focardi et al., 2009; Casale et al., 2012) and decrease to approximately 19°C in October (Balestra et al.,2015). Surface salinities range from 35.2 - 38 in April - May (Campanelli et al., 2013; Monticelli et al., 2014), 37-38 in June (Focardi et al., 2009), and 37.5 in October (Balestra et al., 2015)."**

9) A picture of the mean circulation in the GoM based on HF radar data would be useful, perhaps with 4 panels showing the mean currents during the 4 seasons.
**R: We thank the reviewer for the useful suggestion and in this version of the manuscript we have introduced a new figure showing the Gulf of Manfredonia mean currents and standard deviation for October 2013 and February 2014. We chose these months as they show the highest and lowest residence times of virtual particles in the Gulf. The figure is discussed throughout the text (pg. 6, l. 8-10; pg. 21, l. 13-15).**

10) Have deeper SVP drifters been deployed in the Adriatic? How do trajectories of CODE drifters compare to SVP drifters?
**R: Yes, SVP drifters have been deployed in the Adriatic. However, the SVP dataset consists of only 20 drifters with depth in between 12 and 15m. Compared to the CODE drifters used in this manuscript, the spatial coverage is limited to deeper areas of the basin and no SVP drifters visit the northern Adriatic Sea or the Manfredonia Gulf (Figure 1 of this report). For this reason, only CODE drifters have been used in the analysis presented in this manuscript.**

[Figure]

SVP drifters  (08-Sep-1997 00:00:00 -- 11-Mar-2015 07:00:00)

*Figure1: SVP historical drifters trajectories in the Adriatic*

11) Radar resolution of 1.5 km might be too coarse to fully resolve near-shore processes. Perhaps, a comparison between real and HF-simulated drifters (ensemble- averaged separation between real-simulated drifters as a function of time, or something similar) could be shown to investigate how well the radar represents the actual currents?
**R: The reader is referred to Corgnati et al., (2018) for a complete validation of the Gulf of Manfredonia HF-Radar velocities using in-situ measurements. As stated in Section 3.3, Corgnati et al., 2018 found that the rms (root mean square) of the differences between drifter and HF radar velocities is ~ 20 % - 50 % of the drifter rms velocities, which falls on the lower side of typical errors found in literature (Figure 13 of Corgnati et al., (2018)). In this version of the manuscript we clarified this ambiguity by adding the following sentence (pg. 11, l. 1-2): "The reader is referred to Corgnati et al., 2018 for a detailed description of the HF-Radar validation. "**

12) The 25 m isobath seems a little arbitrary as the outer boundary of the GoM. Would the residence times change a lot if, say, a 30-m isobath is used instead? Same question about the northern and southern boundaries of the bay – would results change drastically if these boundaries are moved a bit?
**R: Compared to other areas, the Gulf of Manfredonia is a well defined bay with the northern and southern boundaries clearly delimited by the Gargano promontory and the abrupt coastline curvature respectively (see Section 2.2).  The gulf differs from the rest of the coast due to the shallow depths**

and gently sloping bottom. At 25 m a thermohaline front and a steep depth gradient ( Fig. 2a) separate inner gulf waters from denser offshore Adriatic Sea waters. For this reason, this isobath has been used in this paper and and in previous works to mark the eastern boundary of the Gulf. While the northern and southern boundaries are geographically well marked the eastern one is somewhat more arbitrary but moving this boundary further offshore doesn't affect the results. In fact, we conducted the same analysis using the 30-m isobath and the residence times of both drifters and virtual particles are qualitative the same. In this version of the manuscript we have included a comment on the sensitivity of our results to the boundary location adding the following sentence: (pg. 21, l. 17;pg. 24, l. 2-4) " **These residence times as well as those computed from surface drifters are robust with respect to the definition of Gulf of Manfredonia. We shifted the eastern boundary offshore (i.e. 30 m isobath) and the results are qualitative the same.**"

**REFERENCES NOT PRESENT IN THE PAPER BUT CITED IN THIS RESPONSE**

Artegiani, A., E. Paschini, A. Russo, D. Bregant, F. Raicich, and N. Pinardi, 1997: The Adriatic Sea General Circulation. Part I: Air–Sea Interactions and Water Mass Structure. J. Phys. Oceanogr., 27, 1492–1514, https://doi.org/10.1175/1520-0485(1997)027<1492:TASGCP>2.0.CO;2

Artegiani, A., E. Paschini, A. Russo, D. Bregant, F. Raicich, and N. Pinardi, 1997: The Adriatic Sea General Circulation. Part II: Baroclinic Circulation Structure. J. Phys. Oceanogr., 27, 1515–1532, https://doi.org/10.1175/1520-0485(1997)027<1515:TASGCP>2.0.CO;2

D'Ortenzio, F., D. Iudicone, C. de Boyer Montegut, P. Testor, D. Antoine, S. Marullo, R. Santoleri, and G. Madec (2005), 
[revised manuscript text omitted]